# Retiring Adult:
# New Datasets for Fair Machine Learning

**Frances Ding**[*]       Moritz Hardt[*]       John Miller[*]            Ludwig Schmidt[*]
UC Berkeley          UC Berkeley         UC Berkeley         Toyota Research Institute

## Abstract

Although the fairness community has recognized the importance of data, researchers in the area primarily rely on UCI Adult when it comes to tabular data. Derived from a 1994 US Census survey, this dataset has appeared in hundreds of research papers where it served as the basis for the development and comparison of many algorithmic fairness interventions. We reconstruct a superset of the UCI Adult data from available US Census sources and reveal idiosyncrasies of the UCI Adult dataset that limit its external validity. Our primary contribution is a suite of new datasets derived from US Census surveys that extend the existing data ecosystem for research on fair machine learning. We create prediction tasks relating to income, employment, health, transportation, and housing. The data span multiple years and all states of the United States, allowing researchers to study temporal shift and geographic variation. We highlight a broad initial sweep of new empirical insights relating to trade-offs between fairness criteria, performance of algorithmic interventions, and the role of distribution shift based on our new datasets. Our findings inform ongoing debates, challenge some existing narratives, and point to future research directions.

## 1 Introduction

Datasets are central to the machine learning ecosystem. Besides providing training and testing data for model builders, datasets formulate problems, organize communities, and interface between academia and industry. Influential works relating to the ethics and fairness of machine learning recognize the centrality of datasets, pointing to significant harms associated with data, as well as better data practices [11, 17, 21, 25, 27]. While the discourse about data has prioritized cognitive domains such as vision, speech, or language, numerous consequential applications of predictive modeling and risk assessment involve bureaucratic, organizational, and administrative records best represented as tabular data [8, 15, 26].

When it comes to tabular data, surprisingly, most research papers on algorithmic fairness continue to involve a fairly limited collection of datasets, chief among them the *UCI Adult* dataset [22]. Derived from the 1994 Current Population Survey conducted by the US Census Bureau, this dataset has made an appearance in more than three hundred research papers related to fairness where it served as the basis for the development and comparison of many algorithmic fairness interventions.

Our work begins with a critical examination of the UCI Adult dataset—its origin, impact, and limitations. To guide this investigation we identify the previously undocumented exact source of the UCI Adult dataset, allowing us to reconstruct a superset of the data from available US Census records.

---

[*]Authors ordered alphabetically
Email: {frances, hardt, miller_john}@berkeley.edu
Email: ludwigschmidt2@gmail.com

35th Conference on Neural Information Processing Systems (NeurIPS 2021).

This reconstruction reveals a significant idiosyncrasy of the UCI Adult prediction task that limits its external validity.

While some issues with UCI Adult are readily apparent, such as its age, limited documentation, and outdated feature encodings, a significant problem may be less obvious at first glance. Specifically, UCI Adult has a binary target label indicating whether the income of a person is greater or less than fifty thousand US dollars. This income threshold of $50k US dollars corresponds to the 76th quantile of individual income in the United States in 1994, the 88th quantile in the Black population, and the 89th quantile among women. We show how empirical findings relating to algorithmic fairness are sensitive to the choice of the income threshold, and how UCI Adult exposes a rather extreme threshold. Specifically, the magnitude of violations in different fairness criteria, trade-offs between them, and the effectiveness of algorithmic interventions all vary significantly with the income threshold. In many cases, the $50k threshold understates and misrepresents the broader picture.

Turning to our primary contribution, we provide a suite of new datasets derived from US Census data that extend the existing data ecosystem for research on fair machine learning. These datasets are derived from two different data products provided by the US Census Bureau. One is the Public Use Microdata Sample of the American Community Survey, involving millions of US households each year. The other is the Annual Social and Economic Supplement of the Current Population Survey. Both released annually, they represent major surveying efforts of the Census Bureau that are the basis of important policy decisions, as well as vital resources for social scientists.

We create prediction tasks in different domains, including income, employment, health, transportation, and housing. The datasets span multiple years and all states of the United States, in particular, allowing researchers to study temporal shift and geographic variation. Alongside these prediction tasks, we release a Python package called `folktables` which interfaces with Census data sources and allows users to both access our new predictions tasks and create new tasks from Census data through a simple API[2].

We contribute a broad initial sweep of new empirical insights into algorithmic fairness based on our new datasets. Our findings inform ongoing debates and in some cases challenge existing narratives about statistical fairness criteria and algorithmic fairness interventions. We highlight three robust observations:

1. Variation within the population plays a major role in empirical observations and how they should be interpreted:

   (a) Fairness criteria and the effect size of different interventions varies greatly by state. This shows that statistical claims about algorithmic fairness must be qualified carefully by context, even though they often are not.

   (b) Training on one state and testing on another generally leads to unpredictable results. Accuracy and fairness criteria could change in either direction. This shows that algorithmic tools developed in one context may not transfer gracefully to another.

   (c) Somewhat surprisingly, fairness criteria appear to be more stable over time than predictive accuracy. This is true both before and after intervention.

2. Algorithmic fairness interventions must specify a locus of intervention. For example, a model could be trained on the entire US population, or on a state-by-state basis. The results differ significantly. Recognition of the need for such a choice is still lacking, as is scholarship guiding the practitioner on how to navigate this choice and its associated trade-offs.

3. Increased dataset size does not necessarily help in reducing observed disparities. Neither does social progress as measured in years passed. This is in contrast to intuition from cognitive machine learning tasks where more representative data can improve metrics such as error rate disparities between different groups.

Our observations apply to years of active research into algorithmic fairness, and our work provides new datasets necessary to re-evaluate and extend the empirical foundations of the field.

---

[2]The datasets and Python package are available for download at `https://github.com/zykls/folktables`.

## 2 Archaeology of UCI Adult: Origin, Impact, Limitations

*Archaeology organises the past to understand the present. It lifts the dust-cover off a world that we take for granted. It makes us reconsider what we experience as inevitable.*

*— Ian Hacking*

Although taken for granted today, the use of benchmark datasets in machine learning emerged only in late 1980s [19]. Created in 1987, the UCI Machine Learning Repository contributed to this development by providing researchers with numerous datasets each with a fixed training and testing split [23]. As of writing, the UCI Adult dataset is the second most popular dataset among more than five hundred datasets in the UCI repository. An identical dataset is called "Census Income Data Set" and a closely related larger dataset goes by "Census-Income (KDD) Data Set".

At the outset, UCI Adult contains 48,842 rows each apparently describing one individual with 14 attributes. The dataset information reveals that it was extracted from the "1994 Census database" according to certain filtering criteria. Since the US Census Bureau provides several data products, as we will review shortly, this piece of information does not identify the source of the dataset.

The fourteen features of UCI Adult include what the fairness community calls *sensitive* or *protected* attributes, such as, age, sex, and race. The earliest paper on algorithmic fairness that used UCI Adult to our knowledge is a work by Calders et al. [12] from 2009. The availability of sensitive attributes contributed to the choice of the dataset for the purposes of this work. An earlier paper in this context by Pedreschi et al. [29] used the UCI German credit dataset, which is smaller and ended up being less widely used in the community. Another highly cited paper on algorithmic fairness that popularized UCI Adult is the work of Zemel et al. [34] on *learning fair representations* (LFR). Published in 2013, the work introduced the idea of changing the data representation to achieve a particular fairness criterion, in this case, demographic parity, while representing the original data as well as possible. This idea remains popular in the community and the LFR method has become a standard baseline.

Representation learning is not the only topic for which UCI Adult became the standard test case. The dataset has become broadly used throughout the area for purposes including the development of new fairness criteria, algorithmic interventions and fairness promoting methods, as well as causal modeling. Major software packages, such as AI Fairness 360 [7] and Fairlearn [9], expose UCI Adult as one of a few standard examples. Indeed, based on bibliographic information available on Google Scholar there appear to be more than 300 papers related to algorithmic fairness that used the UCI Adult dataset at the time of writing.

### 2.1 Reconstruction of UCI Adult

Creating a dataset involves a multitude of design choices that substantially affect the validity of experiments conducted with the dataset. To fully understand the context of UCI Adult and explore variations of its design choices, we reconstructed a closely matching superset from the original Census sources. We now describe our reconstruction in detail and then investigate one specific design choice, the income binarization threshold, in Section 2.2.

The first step in our reconstruction of UCI Adult was identifying the original data source. As mentioned above, the "1994 census database" description in the UCI Adult documentation does not uniquely identify the data product provided by the US Census Bureau. Based on the documentation of the closely related "Census-Income (KDD) Data Set,"[3] we decided to start with the Current Population Survey (CPS) data, specifically the Annual Social and Economic Supplement (ASEC) from 1994. We utilized the IPUMS interface to the CPS data [16] and hence refer to our reconstruction as IPUMS Adult.

The next step in the reconstruction was matching the 15 features in UCI Adult to the CPS data. This was a non-trivial task: the UCI Adult documentation does not mention any specific CPS variable names and IPUMS CPS contains more than 400 candidate variables for the 1994 ASEC. To address this challenge, we designed the following matching procedure that we repeated for each feature in UCI Adult: First, identify a set of candidate variables in CPS via the IPUMS keyword search. For each candidate variable, use the CPS documentation to manually derive a mapping from the CPS

---

[3]Ron Kohavi is a co-creator of both datasets.

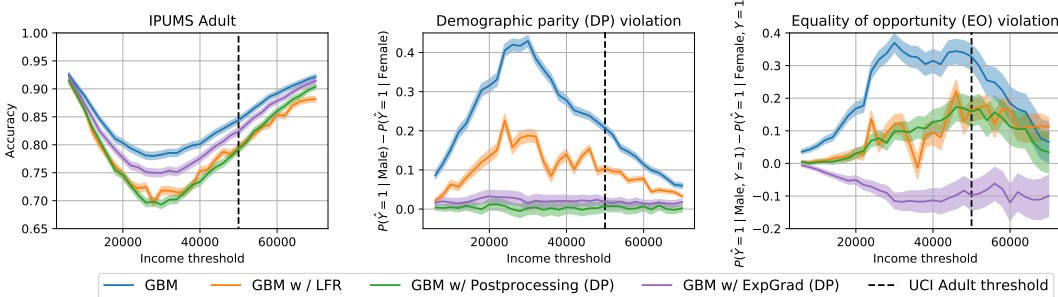

Figure 1: Fairness interventions with varying income threshold on IPUMS Adult. We compare three methods for achieving demographic parity: a pre-processing method (LFR), an in-training method based on Agarwal et al. [2] (ExpGrad), and a post-processing adjustment method [20]. We apply each method using a gradient boosted decision tree (GBM) as the base classifier. Confidence intervals are 95% Clopper-Pearson intervals for accuracy and 95% Newcombe intervals for DP.

encoding to the UCI Adult encoding. Finally, match each row in UCI Adult to its nearest neighbor in the partial reconstruction assembled from previous exact variable matches.

We only included a candidate variable if the nearest neighbor match was *exact*, i.e., we could find an exact match in the IPUMS CPS data for each row in UCI Adult that matched *both* the candidate variable and all earlier variables also identified via exact matches. There were only two exceptions to this rule. We discuss them in Appendix A. After completing the variable matching, our reconstruction has 49,531 rows when we use the same inclusion criteria as UCI Adult to the extent possible, which is slightly more than the 48,842 rows in UCI Adult. The discrepancy likely stems from the fact that UCI Adult used the variable "fnlwgt" in its inclusion criteria and we did not due to the lack of an exact match for this variable. This made our inclusion criteria slightly more permissive than those of UCI Adult. The fact that we found exact matches for 13 of the 15 UCI Adult variables and a very close match for "native-country" is evidence that our reconstruction of UCI Adult is accurate.

## 2.2 Varying income threshold

The goal in the UCI Adult dataset is to predict whether an individual earns greater than 50,000 US dollars a year. The choice of the 50,000 dollar threshold is idiosyncratic and potentially limits the external validity of UCI Adult as a benchmark for algorithmic fairness. In 1994, the median US income was 26,000 dollars, and 50,000 dollars corresponds to the 76th quantile of the income distribution, and the 88th and 89th quantiles of the income distribution for the Black and female populations, respectively. Consequently, *almost all of the Black and female instances in the dataset fall below the threshold* and models trained on UCI adult tend to have substantially higher accuracies on these subpopulations. For instance, a standard logistic regression model trained on UCI Adult dataset achieves 85% accuracy overall, 91.4% accuracy on the Black instances, and 92.7% on Female instances. This is a rather untypical situation since often machine learning models perform more poorly on historically disadvantaged groups.

To understand the sensitivity of the empirical findings on UCI Adult to the choice of threshold, we leverage our IPUMS Adult reconstruction, which includes the continuous, unthresholded income variable, and construct a new collection of datasets where the income threshold varies from 6,000 to 70,000. For each threshold, we first train a standard gradient boosted decision tree and evaluate both its accuracy and its violation of two common fairness criteria: *demographic parity* (equality of positive rates) and *equal opportunity* (equality of true positive rates). See the text by Barocas et al. [6] for background. The results are presented in Figure 1, where we see both accuracy and the magnitude of violations of these criteria vary substantially with the threshold choice.

We then evaluate how the choice of threshold affects three common classes of fairness interventions: the preprocessing method LFR [34] mentioned earlier, an *in-processing* or *in-training* method based on the reductions approach in Agarwal et al. [2], and the post-processing method from Hardt et al. [20]. In Figure 1, we plot model accuracy after applying each intervention to achieve demographic parity as well as violations of both demographic parity and equality of opportunity as the income

Table 1: New prediction task details instantiated on 2018 US-wide ACS PUMS data

| Task | Features | Datapoints | Constant predictor acc | LogReg acc | GBM acc |
|---|---|---|---|---|---|
| ACSIncome | 10 | 1,599,229 | 63.1% | 77.1% | 79.7% |
| ACSPublicCoverage | 19 | 1,127,446 | 70.2% | 75.6% | 78.5 % |
| ACSMobility | 21 | 620,937 | 73.6% | 73.7% | 75.7% |
| ACSEmployment | 17 | 2,320,013 | 56.7% | 74.3% | 78.5% |
| ACSTravelTime | 16 | 1,428,642 | 56.3% | 57.4% | 65.0% |

threshold varies. In Appendix A, we conduct the same experiment for methods to achieve equality of opportunity. There are three salient findings. First, the effectiveness of each intervention depends on the threshold. For values of the threshold near 25,000, the accuracy drop needed to achieve demographic parity or equal opportunity is significantly larger than closer to 50,000. Second, the trade-offs between different criteria vary substantially with the threshold. Indeed, for the in-processing method enforcing demographic parity, as the threshold varies, the equality of opportunity violation is monotonically increasing. Third, for high values of the threshold, the small number of positive instances substantially enlarges the confidence intervals for equality of opportunity, which makes it difficult to meaningfully compare the performance of methods for satisfying this constraint.

## 3 New datasets for algorithmic fairness

At least one aspect of UCI Adult is remarkably positive. The US Census Bureau invests heavily in high quality data collection, surveying methodology, and documentation based on decades of experience. Moreover, responses to some US Census Bureau surveys are legally mandated and hence enjoy high response rates resulting in a representative sample. In contrast, some notable datasets in machine learning are collected in an ad-hoc manner, plagued by skews in representation [10, 13, 32, 33], often lacking copyright [24] or consent from subjects [30], and involving unskilled or poorly compensated labor in the form of crowd workers [18].

In this work, we tap into the vast data ecosystem of the US Census Bureau to create new machine learning tasks that we hope help to establish stronger empirical evaluation practices within the algorithmic fairness community.

As previously discussed, UCI Adult was derived from the Annual Social and Economic Supplement (ASEC) of the Current Population Survey (CPS). The CPS is a monthly survey of approximately 60,000 US households. It's used to produce the official monthly estimates of employment and unemployment for the United States. The ASEC contains additional information collected annually.

Another US Census data product most relevant to us are the American Community Survey (ACS) Public Use Microdata Sample (PUMS). ACS PUMS differs in some significant ways from CPS ASEC. The ACS is sent to approximately 3.5 million US households each year gathering information relating to ancestry, citizenship, education, employment, language proficiency, income, disability, and housing characteristics. Participation in the ACS is mandatory under federal law. Responses are confidential and governed by strict privacy rules. The Public Use Microdata Sample contains responses to every question from a subset of respondents. The geographic information associated with any given record is limited to a level that aims to prevent re-identification of survey participants. A number of other disclosure control heuristics are implemented. Extensive documentation is available on the websites of the US Census Bureau.

### 3.1 Available prediction tasks

We use ACS PUMS as the basis for the following new prediction tasks:

**ACSIncome:** predict whether an individual's income is above $50,000, after filtering the ACS PUMS data sample to only include individuals above the age of 16, who reported usual working hours of at least 1 hour per week in the past year, and an income of at least $100. The threshold of $50,000 was chosen so that this dataset can serve as a replacement to UCI Adult, but we also offer datasets with other income cutoffs described in Appendix B.

**ACSPublicCoverage:** predict whether an individual is covered by public health insurance, after filtering the ACS PUMS data sample to only include individuals under the age of 65, and those with an income of less than $30,000. This filtering focuses the prediction problem on low-income individuals who are not eligible for Medicare.

**ACSMobility:** predict whether an individual had the same residential address one year ago, after filtering the ACS PUMS data sample to only include individuals between the ages of 18 and 35. This filtering increases the difficulty of the prediction task, as the base rate of staying at the same address is above 90% for the general population.

**ACSEmployment:** predict whether an individual is employed, after filtering the ACS PUMS data sample to only include individuals between the ages of 16 and 90.

**ACSTravelTime:** predict whether an individual has a commute to work that is longer than 20 minutes, after filtering the ACS PUMS data sample to only include individuals who are employed and above the age of 16. The threshold of 20 minutes was chosen as it is the US-wide median travel time to work in the 2018 ACS PUMS data release.

All our tasks contain features for age, race, and sex, which correspond to *protected categories* in different domains under US anti-discrimination laws [5]. Further, each prediction task can be instantiated on different ACS PUMS data samples, allowing for comparison across geographic and temporal variation. We provide datasets for each task corresponding to 1) all fifty US states and Puerto Rico, and 2) five different years of data collection: 2014–2018 inclusive, resulting in a total of 255 distinct datasets per task to assess distribution shift. We also provide US-wide datasets for each task, constructed from concatenating each state's data. Table 1 displays more details about each prediction task as instantiated on the 2018 US-wide ACS PUMS data sample. Our new tasks constitute a diverse collection of prediction problems ranging from those where machine learning achieves significantly higher accuracy than a baseline constant predictor to other potentially low-signal problems (ACSMobility) where accuracy improvement appears to be more challenging. We also provide the exact features included in each prediction task, and other details, in Appendix B. A datasheet [17] for our datasets is provided in Appendix E.

These prediction tasks are by no means exhausitive of the potential tasks one can construct using the ACS PUMS data. The `folktables` package we introduce provides a simple API that allows users to construct new tasks using the ACS PUMS data, and we encourage the community to explore additional prediction tasks beyond those introduced in this paper.

## 3.2   Scope and limitations

One distinction is important. Census data is often used by social scientists to study the extent of inequality in income, employment, education, housing or other aspects of life. Such important substantive investigations should necessarily inform debates about discrimination in classification scenarios within these domains. However, our contribution is not in this direction. We instead use census data for the empirical study of algorithmic fairness. This generally may include performance claims about specific methods, the comparison of different methods for achieving a given fairness metric, the relationships of different fairness criteria in concrete settings, causal modeling of different scenarios, and the ability of different methods to transfer successfully from one context to another. We hope that our work leads to more comprehensive empirical evaluations in research papers on the topic, at the very least reducing the overreliance on UCI Adult and providing a complement to the flourishing theoretical work on the topic. The distinction we draw between benchmark data and substantive domain-specific investigations resonates with recent work that points out issues with using data about risk assessments tools from the criminal justice domain as machine learning benchmarks [4].

A notable if obvious limitation of our work is that it is entirely US-centric. A richer dataset ecosystem covering international contexts within the algorithmic fairness community is still lacking. Although empirical work in the Global South is central in other disciplines, there continues to be much need for the North American fairness community to engage with it more strongly [1].

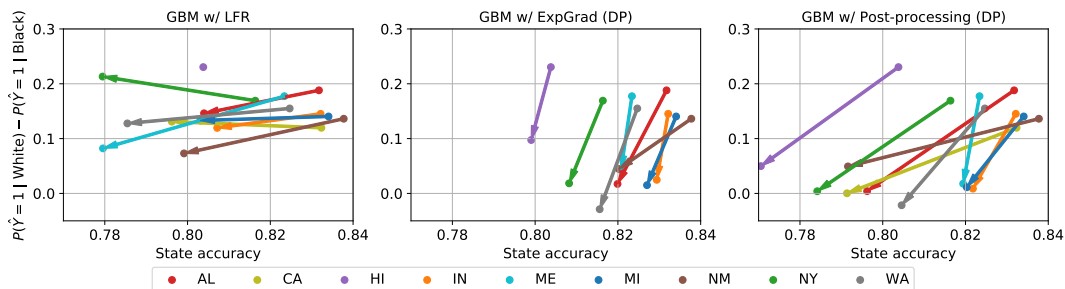

Figure 2: The effect size of fairness interventions varies by state. Each panel shows the change in accuracy and demographic parity on the ACSIncome task after applying a fairness intervention to an unconstrained gradient boosted decision tree (GBM). Each arrow corresponds to a different state distribution. The arrow base represents the (accuracy, DP) point corresponding to the unconstrained GBM, and the head represents the (accuracy, DP) point obtained after applying the intervention. The arrow for HI in the LFR plot is entirely covered by the start and end points.

# 4   A tour of empirical observations

In this section, we highlight an initial sweep of empirical observations enabled by our new ACS PUMS derived prediction tasks. Our experiments focus on three fundamental issues in fair machine learning: (i) variation within the population of interest, e.g., how does the effectiveness of interventions vary between different states or over time?, (ii) the locus of intervention, e.g. should interventions be performed at the state or national level?, and (iii) whether increased dataset size or the passage of time mitigates observed disparities?

Our experiments are not exhaustive and are intended to highlight the perspective a broader empirical evaluation with our new datasets can contribute to addressing questions within algorithmic fairness. The goal of the experiments is not to provide a complete overview of all the questions that one can answer using our datasets. Rather, we hope to inspire other researchers to creatively use our datasets to further probe these question as well as propose new ones leveraging the ACS PUMS data.

## 4.1   Variation within the population

The ACS PUMS prediction tasks present two natural axes of variation: geographic variation between states and temporal variation between years the ACS is conducted. This variation allows us to both measure the performance of different fairness interventions on a broad collection of different distributions, as well as study the performance of these interventions under geographical and temporal *distribution shift* when the test dataset differs from the one on which the model was trained.

Due to space constraints, we focus our experiments in this section on the ACSIncome prediction task with demographic parity as the fairness criterion of interest. We present similar results for our other prediction tasks and fairness criteria, as well as full experimental details in Appendix D.

**Intervention effect sizes vary across states.**   The fifty US states which comprise the ACS PUMS data present a broad set of different experimental conditions on which to evaluate the performance of fairness interventions. At the most basic level, we can train and evaluate different fairness interventions on each of the states and compare the interventions' efficacy on these different distributions. Concretely, we first train an unconstrained gradient boosted decision tree (GBM) on each state, and we compare the accuracy and fairness criterion violation of this unconstrained model with the same model after applying one of three common fairness intervention: pre-processing (LFR), the in-processing fair reductions methods from Agarwal et al. [2] (ExpGrad), and the simple post-processing method that adjusts group-based acceptance thresholds to satisfy a constraint [20]. Figure 2 shows the result of this experiment for the ACSIncome prediction task for interventions to achieve demographic parity. For a given method, performance can differ markedly between states. For instance, LFR decreases the demographic parity violation by 10% in some states and in other states the decrease is close to zero. Similarly, the post-processing adjustment to enforce demographic parity incurs accuracy drops of less than 1% in some states, whereas in others the drop is closer to 5%.

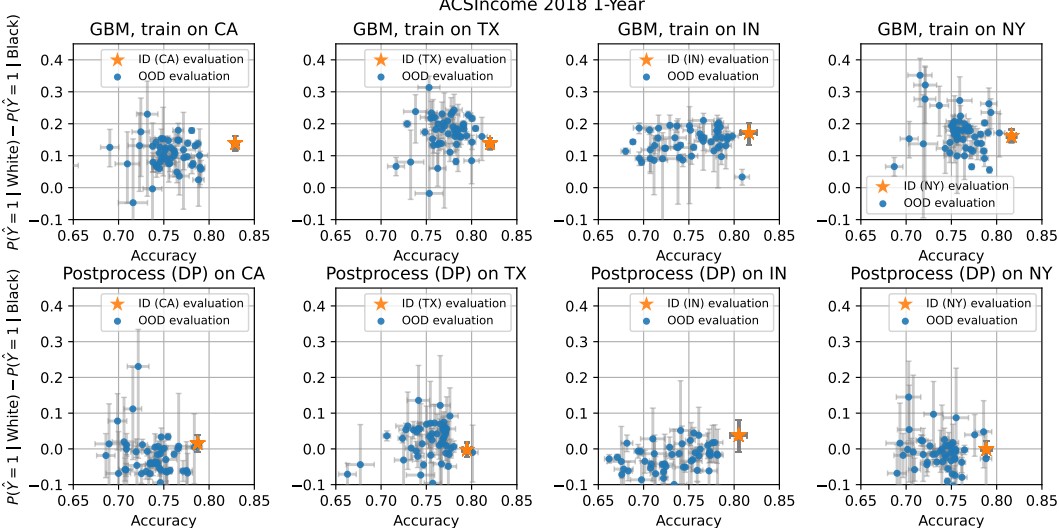

Figure 3: Transfer from one state to another gives unpredictable results in terms of predictive accuracy and fairness criteria. **Top:** Each panel shows an unconstrained GBM trained on a particular state on the ACSIncome task and evaluated both in-distribution (ID) on the same state and out-of-distribution (OOD) on the 49 other states in terms of accuracy and demographic parity violation. **Bottom:** Each panel shows an GBM with post-processing to enforce demographic parity on the state on which it was trained and evaluated both ID and OOD on all 50 states. Confidence intervals are 95% Clopper-Pearson intervals for accuracy and 95% Newcombe intervals for demographic parity.

**Training and testing on different states leads to unpredictable results.** Beyond training and evaluating interventions on different states, we also use the ACS PUMS data to study the performance of interventions under *geographic* distribution shift, where we train a model on one state and test it on another. In Figure 3, we plot accuracy and demographic parity violation with respect to race for both an unconstrained GBM and the same model after applying a post-processing adjustment to achieve demographic parity on a natural suite of test sets: the in-distribution (same state test set) and the out-of-distribution test sets for the 49 other states. For both the unconstrained and post-processed model, model accuracy and demographic parity violation varies substantially across different state test sets. In particular, even when a method achieves demographic parity in one state, it may no longer satisfy the fairness constraint when naively deployed on another.

**Fairness criteria are more stable over time than predictive accuracy.** In contrast to the unpredictable results that occur under geographic distribution shift, the fairness criteria and interventions we study are much more stable under *temporal* distribution shift. Specifically, in Figure 4, we plot model accuracy and demographic parity violation for GBM trained on the ACSIncome task using US-wide data from 2014 and evaluated on the test sets for the same task drawn from years 2014-2018. Perhaps unsurprisingly, model accuracy degrades slightly over time. However, the associated fairness metric is stable and essentially constant over time. Moreover, this same trend holds for the fairness interventions previously discussed. The same base GBM with pre-processing (LFR), in-processing (ExpGrad), or post-processing to satisfy demographic parity in 2014, all have a similar degradation in accuracy, but the fairness metrics remain stable. Thus, a classifier that satisfies demographic parity on the 2014 data continues to satisfy the constraint on 2015-2018 data.

## 4.2 Specifying a locus of intervention

On the ACSPUMs prediction task, fairness interventions can be applied either on a state-by-state basis or on the entire US population. In Table 2, we compare the performance of LFR and the post-processing adjustment method applied at the US-level with the aggregate performance of both methods applied on a state-by-state basis, using a GBM as the base classifier. In both cases, applying the intervention on a state-by-state improves US-wide accuracy while still preserving demographic parity (post-processing) or further mitigating violations of demographic parity (LFR).

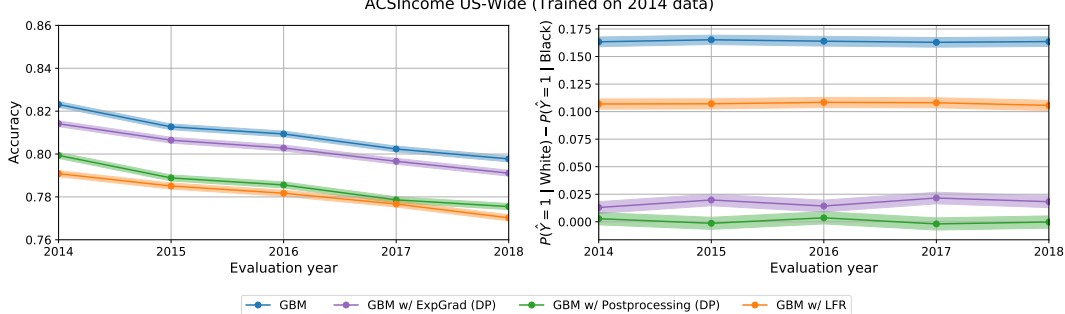

Figure 4: Fairness criteria are more stable over time than accuracy. **Left:** Models trained in 2014 on US-wide ACSIncome with and without fairness interventions to achieve demographic parity and evaluated on data in subsequent years suffer a drop in accuracy over time. **Right:** However, the violation of demographic parity remains essentially constant over time. Confidence intervals are 95% Clopper-Pearson intervals for accuracy and 95% Newcombe intervals for demographic parity.

Table 2: Comparison of two different strategies for applying an intervention to achieve demographic parity (DP) on the US-wide ACSIncome task. *US-level* corresponds to training one classifier and applying the intervention on the entire US population. *State-level* corresponds to training a classifier and applying the intervention separately for each state and then aggregating the results over all states. Here, DP refers to $P(\hat{Y} = 1 \mid \text{White}) - P(\hat{Y} = 1 \mid \text{Black})$. Confidence intervals are 95% Clopper-Pearson intervals for accuracy and 95% Newcombe intervals for DP.

|  | US-level acc | US-level DP violation | State-level acc | State-level DP violation |
|---|---|---|---|---|
| Unconstrained GBM | $81.7 \pm 0.1\,\%$ | $17.7 \pm 0.2\%$ | $82.8 \pm 0.1\,\%$ | $16.9 \pm 0.2\%$ |
| GBM w/ LFR | $78.7 \pm 0.1\,\%$ | $16.6 \pm 0.2\%$ | $79.4 \pm 0.1\%$ | $14.0 \pm 0.2\%$ |
| GBM w/ post-processing (DP) | $79.2 \pm 0.1\,\%$ | $0.3 \pm 0.3\,\%$ | $80.2 \pm 0.1\%$ | $-0.6 \pm 0.3\%$ |

### 4.3 Increased dataset size doesn't necessarily mitigate observed disparities

To mitigate disparities in error rates, commonly suggested remedies include collecting a) larger datasets and b) more representative data reflective of social progress. For example, in response to research revealing the stark accuracy disparities of commercial facial recognition algorithms, particularly for dark-skinned females [11], IBM collected a more diverse training set of images, retrained its facial recognition model, and reported a 10-fold decrease in error for this subgroup [31]. However, on our tabular datasets, larger datasets collected in more socially progressive times do not automatically mitigate disparities. Table 3 shows that unconstrained gradient boosted decision tree trained on a newer, larger dataset (ACSIncome vs. IPUMS Adult), does not improve disparities such as in true positive rate (TPR). A fundamental reason for this is the persistent social inequality that is reflected in the data. It is well known that given a disparity in base rates between groups, a predictive model cannot be both calibrated and equal in error rates across groups [14], except if the model has 100% accuracy. This observation highlights a key difference between cognitive machine learning and tabular data prediction – the Bayes error rate is zero for cognitive machine learning. Thus larger and more representative datasets eventually address disparities by pushing error rates to zero for all subgroups. In the tabular datasets we collect, the Bayes error rate of an optimal classifier is almost certainly far from zero, so some individuals will inevitably be incorrectly classified. Rather than hope for future datasets to implicitly address disparities, we must directly contend with how dataset and model design choices distribute the burden of these errors.

## 5 Discussion and future directions

Rather than settled conclusions, our empirical observations are intended to spark additional work on our new datasets. Of particular interest is a broad and comprehensive evaluation of existing methods on all datasets. We only evaluated some methods so far. One interesting question is if there is a method

Table 3: Disparities persist despite increasing dataset size and social progress.

| Dataset | Year | Datapoints | GBM acc | TPR White | TPR Black | TPR disparity |
|---|---|---|---|---|---|---|
| IPUMS Adult | 1994 | 49,531 | 86.4% | 58.0% | 46.5 % | 11.5% |
| ACSIncome | 2018 | 1,599,229 | 80.8% | 66.5% | 51.7% | 14.8% |

for achieving either demographic parity or error rate parity that outperforms threshold adjustment (based on the best known unconstrained classifier) on any of our datasets? We conjecture that the answer is *no*. The reason is that we believe on our datasets a well-tuned tree-ensemble achieves classification error close to the Bayes error bound. Existing theory (Theorem 5.3 in [20]) would then show that threshold adjustment based on this model is, in fact, optimal. Our conjecture motivates drawing a distinction between classification scenarios where a nearly Bayes optimal classifier is known and those where there isn't. How close we are to Bayes optimal on any of our new prediction tasks is a good question. The role of distribution shift also deserves more attention. Are there methods that achieve consistent performance across geographic contexts? Why does there appear to be more temporal than geographic stability? What does the sensitivity to distribution shift say about algorithmic tools developed in one context and deployed in another? Answers to these questions seem highly relevant to policy-making around the deployment of algorithmic risk assessment tools. Finally, our datasets are also interesting test cases for causal inference methods, which we haven't yet explored. How would, for example, methods like *invariant risk minimization* [3] perform on different geographic contexts?

## Acknowledgements

We thank Barry Becker and Ronny Kohavi for answering our many questions around the origin and creation of the UCI Adult dataset. FD and JM are supported by the National Science Foundation Graduate Research Fellowship Program under Grant No. DGE 1752814. FD is additionally supported by the Open Philanthropy Project AI Fellows Program.

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
