# Contents

# A Adult reconstruction

## A.1 Additional reconstruction details

We only included a candidate variable if the nearest neighbor match was *exact*, i.e., we could find an exact match in the IPUMS CPS data for each row in UCI Adult that matched *both* the candidate variable and all earlier variables also identified via exact matches. There were only two exceptions to this rule:

- The UCI Adult feature "native-country". Here we could match the vast majority of rows in UCI Adult to the IPUMS CPS variable "UH_NATVTY_A1". To get an exact match for all rows, we had to map the country codes for Russia and Guyana in "UH_NATVTY_A1" to the value for "unknown". The documentation for UCI Adult also mentions neither Russia nor Guyana as possible values for "native-country". We do not know the reason for this discrepancy.

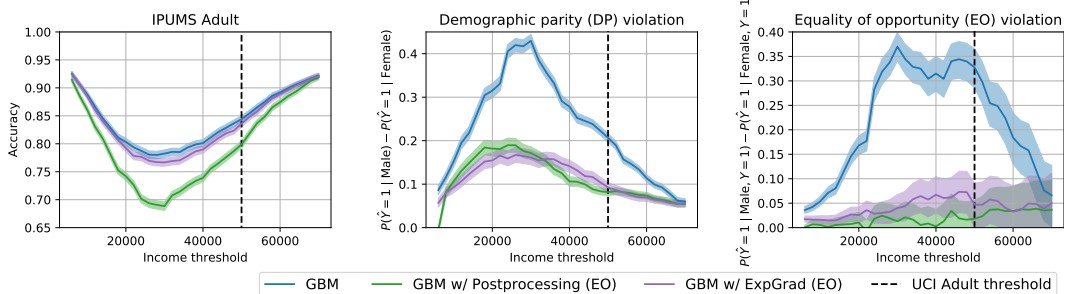

Figure 5: Fairness interventions with varying income threshold on IPUMS Adult. Comparison of in-processing and post-processing methods for achieving equality of opportunity (EO). LFR does not target EO, so we exclude it from the comparison. Confidence intervals are 95% Clopper-Pearson intervals for accuracy and 95% Newcombe intervals for equality of opportunity.

- The UCI Adult feature "fnlwgt". This column is actually not a demographic feature of an individual but a weight value computed by the Census Bureau to make the sample representative for the US population. We compared the "fnlwgt" data to all weight variables available in IPUMS CPS but did not find an exact match. The closest match is the variable "UH_WGTS_A1", which has a similar distribution. Since we did not identify an exact match for "fnlwgt" and the variable is not a property of an individual, we do not utilize it further in our experiments.

## A.2   Varying the income threshold experiments

In our experiments, we randomly split the 49,531 examples in the IPUMS Adult reconstruction into a training set of size 32,094 and a test-set of size 13,755. We vary the threshold from 6,000 to 72,000. Concretely, for a given threshold, e.g. 25,000, the task is to predict whether the individual's income is greater than 25,000. We use a one-hot encoding for the categorical features, and we use the same clustering preprocessing for the `Education-Num` and `Age` features as Bellamy et al. [7]. All features are further scaled to be zero-mean and have unit variance.

In our experiments, as the "unconstrained" base classifier, we use the gradient boosted decision tree classifier provided by Pedregosa et al. [28] with exponential loss, `num_estimators` 5, `max_depth` 5, and all other hyperparameters set to the default. We found this to slightly outperform the default gradient boosting machine at threshold 50,000. For the three fairness interventions, we used the implementation of LFR [34] provided by Bellamy et al. [7] with hyperparameters `Ax` 1e-4, `Ay` 1.0, `Az` 1000, `maxiter` 20000, and `maxfun` 20000, which were chosen by a grid search at threshold 50,000 to maximize the difference between accuracy and the demographic parity disparity. We used the implementation of the reductions approach of Agarwal et al. [2] provided by Bird et al. [9] with the default hyperparameters, and we used implementation of post-processing [20] provided by Bellamy et al. [7].

In Figure 1 in the main text, we compare the performance of these three fairness interventions when enforcing demographic parity as the threshold varies. In Figure 5, we additionally compare the performance of in-processing method (ExpGrad) and the post-processing method when enforcing equality of opportunity (EO). We exclude LFR from the comparison because this method does not enforce equality of opportunity without additional modification. The results from this experiment are very similar to the experiment enforcing demographic parity. As the threshold varies, the accuracy drop needed to enforce EO varies substantially, as does the trade-off between criteria when enforcing EO. Moreover, for high values of the threshold, the small number of positive instances substantially increases the confidence intervals around the report EO values and makes it difficult to compare the different interventions.

# B    New prediction task details

In this section we detail the target variable, features, and filters that comprise each of our prediction tasks; more information about each feature can be found from the ACS PUMS documentation.[4] For each feature, we list the variable code as provided by the ACS PUMS data sample, its extended description in parentheses, and finally the range of values for the variable.

## B.1    ACSIncome

Predict whether US working adults' yearly income is above $50,000.

**Target:**   PINCP (Total person's income): an individual's label is 1 if PINCP > 50000, otherwise 0. Note that with our software package, this chosen income threshold can be toggled easily to label the ACS PUMS data differently, and construct a new prediction task.

**Features:**

- AGEP (Age): Range of values:
  - 0 - 99 (integers)
  - 0 indicates less than 1 year old.
- COW (Class of worker): Range of values:
  - N/A (not in universe)
  - 1: Employee of a private for-profit company or business, or of an individual, for wages, salary, or commissions
  - 2: Employee of a private not-for-profit, tax-exempt, or charitable organization
  - 3: Local government employee (city, county, etc.)
  - 4: State government employee
  - 5: Federal government employee
  - 6: Self-employed in own not incorporated business, professional practice, or farm
  - 7: Self-employed in own incorporated business, professional practice or farm
  - 8: Working without pay in family business or farm
  - 9: Unemployed and last worked 5 years ago or earlier or never worked
- SCHL (Educational attainment): Range of values:
  - N/A (less than 3 years old)
  - 1: No schooling completed
  - 2: Nursery school/preschool
  - 3: Kindergarten
  - 4: Grade 1
  - 5: Grade 2
  - 6: Grade 3
  - 7: Grade 4
  - 8: Grade 5
  - 9: Grade 6
  - 10: Grade 7
  - 11: Grade 8
  - 12: Grade 9
  - 13: Grade 10
  - 14: Grade 11
  - 15: 12th Grade - no diploma
  - 16: Regular high school diploma
  - 17: GED or alternative credential

---

[4]https://www.census.gov/programs-surveys/acs/microdata/documentation.html

- 18: Some college but less than 1 year
- 19: 1 or more years of college credit but no degree
- 20: Associate's degree
- 21: Bachelor's degree
- 22: Master's degree
- 23: Professional degree beyond a bachelor's degree
- 24: Doctorate degree

- MAR (Marital status): Range of values:
  - 1: Married
  - 2: Widowed
  - 3: Divorced
  - 4: Separated
  - 5: Never married or under 15 years old

- OCCP (Occupation): Please see ACS PUMS documentation for the full list of occupation codes

- POBP (Place of birth): Range of values includes most countries and individual US states; please see ACS PUMS documentation for the full list.

- RELP (Relationship): Range of values:
  - 0: Reference person
  - 1: Husband/wife
  - 2: Biological son or daughter
  - 3: Adopted son or daughter
  - 4: Stepson or stepdaughter
  - 5: Brother or sister
  - 6: Father or mother
  - 7: Grandchild
  - 8: Parent-in-law
  - 9: Son-in-law or daughter-in-law
  - 10: Other relative
  - 11: Roomer or boarder
  - 12: Housemate or roommate
  - 13: Unmarried partner
  - 14: Foster child
  - 15: Other nonrelative
  - 16: Institutionalized group quarters population
  - 17: Noninstitutionalized group quarters population

- WKHP (Usual hours worked per week past 12 months): Range of values:
  - N/A (less than 16 years old / did not work during the past 12 months)
  - 1 - 98 integer valued: usual hours worked
  - 99: 99 or more usual hours

- SEX (Sex): Range of values:
  - 1: Male
  - 2: Female

- RAC1P (Recoded detailed race code): Range of values:
  - 1: White alone
  - 2: Black or African American alone
  - 3: American Indian alone
  - 4: Alaska Native alone

- 5: American Indian and Alaska Native tribes specified, or American Indian or Alaska Native, not specified and no other races
- 6: Asian alone
- 7: Native Hawaiian and Other Pacific Islander alone
- 8: Some Other Race alone
- 9: Two or More Races

**Filters:**

- AGEP (Age): Must be greater than 16
- PINCP (Total person's income): Must be greater than 100
- WKHP (Usual hours worked per week past 12 months): Must be greater than 0
- PWGTP (Person weight (relevant for re-weighting dataset to represent the general US population most accurately)): Must be greater than or equal to 1

## B.2 ACSPublicCoverage

Predict whether a low-income individual, not eligible for Medicare, has coverage from public health insurance.

**Target:** PUBCOV (Public health coverage): an individual's label is 1 if PUBCOV == 1 (with public health coverage), otherwise 0.

**Features:**

- AGEP (Age): Range of values:
  - 0 - 99 (integers)
  - 0 indicates less than 1 year old.
- SCHL (Educational attainment): Range of values:
  - N/A (less than 3 years old)
  - 1: No schooling completed
  - 2: Nursery school/preschool
  - 3: Kindergarten
  - 4: Grade 1
  - 5: Grade 2
  - 6: Grade 3
  - 7: Grade 4
  - 8: Grade 5
  - 9: Grade 6
  - 10: Grade 7
  - 11: Grade 8
  - 12: Grade 9
  - 13: Grade 10
  - 14: Grade 11
  - 15: 12th Grade - no diploma
  - 16: Regular high school diploma
  - 17: GED or alternative credential
  - 18: Some college but less than 1 year
  - 19: 1 or more years of college credit but no degree
  - 20: Associate's degree
  - 21: Bachelor's degree
  - 22: Master's degree

- – 23: Professional degree beyond a bachelor's degree
- – 24: Doctorate degree
- MAR (Marital status): Range of values:
  - – 1: Married
  - – 2: Widowed
  - – 3: Divorced
  - – 4: Separated
  - – 5: Never married or under 15 years old
- SEX (Sex): Range of values:
  - – 1: Male
  - – 2: Female
- DIS (Disability recode): Range of values:
  - – 1: With a disability
  - – 2: Without a disability
- ESP (Employment status of parents): Range of values:
  - – N/A (not own child of householder, and not child in subfamily)
  - – 1: Living with two parents: both parents in labor force
  - – 2: Living with two parents: Father only in labor force
  - – 3: Living with two parents: Mother only in labor force
  - – 4: Living with two parents: Neither parent in labor force
  - – 5: Living with father: Father in the labor force
  - – 6: Living with father: Father not in labor force
  - – 7: Living with mother: Mother in the labor force
  - – 8: Living with mother: Mother not in labor force
- CIT (Citizenship status): Range of values:
  - – 1: Born in the U.S.
  - – 2: Born in Puerto Rico, Guam, the U.S. Virgin Islands, or the Northern Marianas
  - – 3: Born abroad of American parent(s)
  - – 4: U.S. citizen by naturalization
  - – 5: Not a citizen of the U.S.
- MIG (Mobility status (lived here 1 year ago)): Range of values:
  - – N/A (less than 1 year old)
  - – 1: Yes, same house (nonmovers)
  - – 2: No, outside US and Puerto Rico
  - – 3: No, different house in US or Puerto Rico
- MIL (Military service): Range of values:
  - – N/A (less than 17 years old)
  - – 1: Now on active duty
  - – 2: On active duty in the past, but not now
  - – 3: Only on active duty for training in Reserves/National Guard
  - – 4: Never served in the military
- ANC (Ancestry recode): Range of values:
  - – 1: Single
  - – 2: Multiple
  - – 3: Unclassified
  - – 4: Not reported
  - – 8: Suppressed for data year 2018 for select PUMAs
- NATIVITY (Nativity): Range of values:

- 1: Native
- 2: Foreign born
- DEAR (Hearing difficulty): Range of values:
  - 1: Yes
  - 2: No
- DEYE (Vision difficulty): Range of values:
  - 1: Yes
  - 2: No
- DREM (Cognitive difficulty): Range of values:
  - N/A (less than 5 years old)
  - 1: Yes
  - 2: No
- PINCP (Total person's income): Range of values:
  - integers between -19997 and 4209995 to indicate income in US dollars
  - loss of $19998 or more is coded as -19998.
  - income of $4209995 or more is coded as 4209995.
- ESR (Employment status recode): Range of values:
  - N/A (less than 16 years old)
  - 1: Civilian employed, at work
  - 2: Civilian employed, with a job but not at work
  - 3: Unemployed
  - 4: Armed forces, at work
  - 5: Armed forces, with a job but not at work
  - 6: Not in labor force
- ST (State code): Please see ACS PUMS documentation for the correspondence between coded values and state name.
- FER (Gave birth to child within the past 12 months): Range of values:
  - N/A (less than 15 years/greater than 50 years/male)
  - 1: Yes
  - 2: No
- RAC1P (Recoded detailed race code): Range of values:
  - 1: White alone
  - 2: Black or African American alone
  - 3: American Indian alone
  - 4: Alaska Native alone
  - 5: American Indian and Alaska Native tribes specified, or American Indian or Alaska Native, not specified and no other races
  - 6: Asian alone
  - 7: Native Hawaiian and Other Pacific Islander alone
  - 8: Some Other Race alone
  - 9: Two or More Races

**Filters:**

- AGEP (Age) must be less than 65.
- PINCP (Total person's income) must be less than $30,000.

### B.3   ACSMobility

Predict whether a young adult moved addresses in the last year.

**Target:** MIG (Mobility status): an individual's label is 1 if MIG == 1, and 0 otherwise.

**Features:**

- AGEP (Age): Range of values:
  - 0 - 99 (integers)
  - 0 indicates less than 1 year old.
- SCHL (Educational attainment): Range of values:
  - N/A (less than 3 years old)
  - 1: No schooling completed
  - 2: Nursery school/preschool
  - 3: Kindergarten
  - 4: Grade 1
  - 5: Grade 2
  - 6: Grade 3
  - 7: Grade 4
  - 8: Grade 5
  - 9: Grade 6
  - 10: Grade 7
  - 11: Grade 8
  - 12: Grade 9
  - 13: Grade 10
  - 14: Grade 11
  - 15: 12th Grade - no diploma
  - 16: Regular high school diploma
  - 17: GED or alternative credential
  - 18: Some college but less than 1 year
  - 19: 1 or more years of college credit but no degree
  - 20: Associate's degree
  - 21: Bachelor's degree
  - 22: Master's degree
  - 23: Professional degree beyond a bachelor's degree
  - 24: Doctorate degree
- MAR (Marital status): Range of values:
  - 1: Married
  - 2: Widowed
  - 3: Divorced
  - 4: Separated
  - 5: Never married or under 15 years old
- SEX (Sex): Range of values:
  - 1: Male
  - 2: Female
- DIS (Disability recode): Range of values:
  - 1: With a disability
  - 2: Without a disability
- ESP (Employment status of parents): Range of values:
  - N/A (not own child of householder, and not child in subfamily)
  - 1: Living with two parents: both parents in labor force
  - 2: Living with two parents: Father only in labor force
  - 3: Living with two parents: Mother only in labor force

- 4: Living with two parents: Neither parent in labor force
  - 5: Living with father: Father in the labor force
  - 6: Living with father: Father not in labor force
  - 7: Living with mother: Mother in the labor force
  - 8: Living with mother: Mother not in labor force
- CIT (Citizenship status): Range of values:
  - 1: Born in the U.S.
  - 2: Born in Puerto Rico, Guam, the U.S. Virgin Islands, or the Northern Marianas
  - 3: Born abroad of American parent(s)
  - 4: U.S. citizen by naturalization
  - 5: Not a citizen of the U.S.
- MIL (Military service): Range of values:
  - N/A (less than 17 years old)
  - 1: Now on active duty
  - 2: On active duty in the past, but not now
  - 3: Only on active duty for training in Reserves/National Guard
  - 4: Never served in the military
- ANC (Ancestry recode): Range of values:
  - 1: Single
  - 2: Multiple
  - 3: Unclassified
  - 4: Not reported
  - 8: Suppressed for data year 2018 for select PUMAs
- NATIVITY (Nativity): Range of values:
  - 1: Native
  - 2: Foreign born
- RELP (Relationship): Range of values:
  - 0: Reference person
  - 1: Husband/wife
  - 2: Biological son or daughter
  - 3: Adopted son or daughter
  - 4: Stepson or stepdaughter
  - 5: Brother or sister
  - 6: Father or mother
  - 7: Grandchild
  - 8: Parent-in-law
  - 9: Son-in-law or daughter-in-law
  - 10: Other relative
  - 11: Roomer or boarder
  - 12: Housemate or roommate
  - 13: Unmarried partner
  - 14: Foster child
  - 15: Other nonrelative
  - 16: Institutionalized group quarters population
  - 17: Noninstitutionalized group quarters population
- DEAR (Hearing difficulty): Range of values:
  - 1: Yes
  - 2: No
- DEYE (Vision difficulty): Range of values:

- 1: Yes
- 2: No

- DREM (Cognitive difficulty): Range of values:
  - N/A (less than 5 years old)
  - 1: Yes
  - 2: No

- RAC1P (Recoded detailed race code): Range of values:
  - 1: White alone
  - 2: Black or African American alone
  - 3: American Indian alone
  - 4: Alaska Native alone
  - 5: American Indian and Alaska Native tribes specified, or American Indian or Alaska Native, not specified and no other races
  - 6: Asian alone
  - 7: Native Hawaiian and Other Pacific Islander alone
  - 8: Some Other Race alone
  - 9: Two or More Races

- GCL (Grandparents living with grandchildren): Range of values:
  - N/A (less than 30 years/institutional GQ)
  - 1: Yes
  - 2: No

- COW (Class of worker): Range of values:
  - N/A (not in universe)
  - 1: Employee of a private for-profit company or business, or of an individual, for wages, salary, or commissions
  - 2: Employee of a private not-for-profit, tax-exempt, or charitable organization
  - 3: Local government employee (city, county, etc.)
  - 4: State government employee
  - 5: Federal government employee
  - 6: Self-employed in own not incorporated business, professional practice, or farm
  - 7: Self-employed in own incorporated business, professional practice or farm
  - 8: Working without pay in family business or farm
  - 9: Unemployed and last worked 5 years ago or earlier or never worked

- ESR (Employment status recode): Range of values:
  - N/A (less than 16 years old)
  - 1: Civilian employed, at work
  - 2: Civilian employed, with a job but not at work
  - 3: Unemployed
  - 4: Armed forces, at work
  - 5: Armed forces, with a job but not at work
  - 6: Not in labor force

- WKHP (Usual hours worked per week past 12 months): Range of values:
  - N/A (less than 16 years old / did not work during the past 12 months)
  - 1 - 98 integer valued: usual hours worked
  - 99: 99 or more usual hours

- JWMNP (Travel time to work): Range of values:
  - N/A (not a worker or a worker that worked at home)
  - integers 1 - 200 for minutes to get to work
  - top-coded at 200 so values above 200 are coded as 200

- PINCP (Total person's income): Range of values:
    - integers between -19997 and 4209995 to indicate income in US dollars
    - loss of $19998 or more is coded as -19998.
    - income of $4209995 or more is coded as 4209995.

**Filters:**

- AGEP (Age) must be greater than 18 and less than 35.

## B.4 ACSEmployment

Predict whether an adult is employed.

**Target:** ESR (Employment status recode): an individual's label is 1 if ESR == 1, and 0 otherwise.

**Features:**

- AGEP (Age): Range of values:
    - 0 - 99 (integers)
    - 0 indicates less than 1 year old.
- SCHL (Educational attainment): Range of values:
    - N/A (less than 3 years old)
    - 1: No schooling completed
    - 2: Nursery school/preschool
    - 3: Kindergarten
    - 4: Grade 1
    - 5: Grade 2
    - 6: Grade 3
    - 7: Grade 4
    - 8: Grade 5
    - 9: Grade 6
    - 10: Grade 7
    - 11: Grade 8
    - 12: Grade 9
    - 13: Grade 10
    - 14: Grade 11
    - 15: 12th Grade - no diploma
    - 16: Regular high school diploma
    - 17: GED or alternative credential
    - 18: Some college but less than 1 year
    - 19: 1 or more years of college credit but no degree
    - 20: Associate's degree
    - 21: Bachelor's degree
    - 22: Master's degree
    - 23: Professional degree beyond a bachelor's degree
    - 24: Doctorate degree
- MAR (Marital status): Range of values:
    - 1: Married
    - 2: Widowed
    - 3: Divorced
    - 4: Separated
    - 5: Never married or under 15 years old

- SEX (Sex): Range of values:
  - 1: Male
  - 2: Female
- DIS (Disability recode): Range of values:
  - 1: With a disability
  - 2: Without a disability
- ESP (Employment status of parents): Range of values:
  - N/A (not own child of householder, and not child in subfamily)
  - 1: Living with two parents: both parents in labor force
  - 2: Living with two parents: Father only in labor force
  - 3: Living with two parents: Mother only in labor force
  - 4: Living with two parents: Neither parent in labor force
  - 5: Living with father: Father in the labor force
  - 6: Living with father: Father not in labor force
  - 7: Living with mother: Mother in the labor force
  - 8: Living with mother: Mother not in labor force
- MIG (Mobility status (lived here 1 year ago): Range of values:
  - N/A (less than 1 year old)
  - 1: Yes, same house (nonmovers)
  - 2: No, outside US and Puerto Rico
  - 3: No, different house in US or Puerto Rico
- CIT (Citizenship status): Range of values:
  - 1: Born in the U.S.
  - 2: Born in Puerto Rico, Guam, the U.S. Virgin Islands, or the Northern Marianas
  - 3: Born abroad of American parent(s)
  - 4: U.S. citizen by naturalization
  - 5: Not a citizen of the U.S.
- MIL (Military service): Range of values:
  - N/A (less than 17 years old)
  - 1: Now on active duty
  - 2: On active duty in the past, but not now
  - 3: Only on active duty for training in Reserves/National Guard
  - 4: Never served in the military
- ANC (Ancestry recode): Range of values:
  - 1: Single
  - 2: Multiple
  - 3: Unclassified
  - 4: Not reported
  - 8: Suppressed for data year 2018 for select PUMAs
- NATIVITY (Nativity): Range of values:
  - 1: Native
  - 2: Foreign born
- RELP (Relationship): Range of values:
  - 0: Reference person
  - 1: Husband/wife
  - 2: Biological son or daughter
  - 3: Adopted son or daughter
  - 4: Stepson or stepdaughter

- 5: Brother or sister
- 6: Father or mother
- 7: Grandchild
- 8: Parent-in-law
- 9: Son-in-law or daughter-in-law
- 10: Other relative
- 11: Roomer or boarder
- 12: Housemate or roommate
- 13: Unmarried partner
- 14: Foster child
- 15: Other nonrelative
- 16: Institutionalized group quarters population
- 17: Noninstitutionalized group quarters population

- DEAR (Hearing difficulty): Range of values:
  - 1: Yes
  - 2: No

- DEYE (Vision difficulty): Range of values:
  - 1: Yes
  - 2: No

- DREM (Cognitive difficulty): Range of values:
  - N/A (less than 5 years old)
  - 1: Yes
  - 2: No

- RAC1P (Recoded detailed race code): Range of values:
  - 1: White alone
  - 2: Black or African American alone
  - 3: American Indian alone
  - 4: Alaska Native alone
  - 5: American Indian and Alaska Native tribes specified, or American Indian or Alaska Native, not specified and no other races
  - 6: Asian alone
  - 7: Native Hawaiian and Other Pacific Islander alone
  - 8: Some Other Race alone
  - 9: Two or More Races

- GCL (Grandparents living with grandchildren): Range of values:
  - N/A (less than 30 years/institutional GQ)
  - 1: Yes
  - 2: No

**Filters:**

- AGEP (Age) must be greater than 16 and less than 90.
- PWGTP (Person weight) must be greater than or equal to 1.

## B.5 ACSTravelTime

Predict whether a working adult has a travel time to work of greater than 20 minutes.

**Target:** JWMNP (Travel time to work): an individual's label is 1 if JWMNP > 20, and 0 otherwise.

**Features:**

- AGEP (Age): Range of values:
  - 0 - 99 (integers)
  - 0 indicates less than 1 year old.
- SCHL (Educational attainment): Range of values:
  - N/A (less than 3 years old)
  - 1: No schooling completed
  - 2: Nursery school/preschool
  - 3: Kindergarten
  - 4: Grade 1
  - 5: Grade 2
  - 6: Grade 3
  - 7: Grade 4
  - 8: Grade 5
  - 9: Grade 6
  - 10: Grade 7
  - 11: Grade 8
  - 12: Grade 9
  - 13: Grade 10
  - 14: Grade 11
  - 15: 12th Grade - no diploma
  - 16: Regular high school diploma
  - 17: GED or alternative credential
  - 18: Some college but less than 1 year
  - 19: 1 or more years of college credit but no degree
  - 20: Associate's degree
  - 21: Bachelor's degree
  - 22: Master's degree
  - 23: Professional degree beyond a bachelor's degree
  - 24: Doctorate degree
- MAR (Marital status): Range of values:
  - 1: Married
  - 2: Widowed
  - 3: Divorced
  - 4: Separated
  - 5: Never married or under 15 years old
- SEX (Sex): Range of values:
  - 1: Male
  - 2: Female
- DIS (Disability recode): Range of values:
  - 1: With a disability
  - 2: Without a disability
- ESP (Employment status of parents): Range of values:
  - N/A (not own child of householder, and not child in subfamily)
  - 1: Living with two parents: both parents in labor force
  - 2: Living with two parents: Father only in labor force
  - 3: Living with two parents: Mother only in labor force
  - 4: Living with two parents: Neither parent in labor force

- 5: Living with father: Father in the labor force
- 6: Living with father: Father not in labor force
- 7: Living with mother: Mother in the labor force
- 8: Living with mother: Mother not in labor force

- MIG (Mobility status (lived here 1 year ago)): Range of values:
  - N/A (less than 1 year old)
  - 1: Yes, same house (nonmovers)
  - 2: No, outside US and Puerto Rico
  - 3: No, different house in US or Puerto Rico

- RELP (Relationship): Range of values:
  - 0: Reference person
  - 1: Husband/wife
  - 2: Biological son or daughter
  - 3: Adopted son or daughter
  - 4: Stepson or stepdaughter
  - 5: Brother or sister
  - 6: Father or mother
  - 7: Grandchild
  - 8: Parent-in-law
  - 9: Son-in-law or daughter-in-law
  - 10: Other relative
  - 11: Roomer or boarder
  - 12: Housemate or roommate
  - 13: Unmarried partner
  - 14: Foster child
  - 15: Other nonrelative
  - 16: Institutionalized group quarters population
  - 17: Noninstitutionalized group quarters population

- RAC1P (Recoded detailed race code): Range of values:
  - 1: White alone
  - 2: Black or African American alone
  - 3: American Indian alone
  - 4: Alaska Native alone
  - 5: American Indian and Alaska Native tribes specified, or American Indian or Alaska Native, not specified and no other races
  - 6: Asian alone
  - 7: Native Hawaiian and Other Pacific Islander alone
  - 8: Some Other Race alone
  - 9: Two or More Races

- PUMA (Public use microdata area code (PUMA) based on 2010 Census definition (areas with population of 100,000 or more, use with ST for unique code)): Please see ACS PUMS documentation for details on the PUMA codes (which range from 100 to 70301)

- ST (State code): Please see ACS PUMS documentation for the correspondence between coded values and state name.

- CIT (Citizenship status): Range of values:
  - 1: Born in the U.S.
  - 2: Born in Puerto Rico, Guam, the U.S. Virgin Islands, or the Northern Marianas
  - 3: Born abroad of American parent(s)
  - 4: U.S. citizen by naturalization
  - 5: Not a citizen of the U.S.

- OCCP (Occupation): Please see ACS PUMS documentation for the full list of occupation codes
- JWTR (Means of transportation to work): Range of values:
  - N/A (not a worker–not in the labor force, including persons under 16 years, unemployed, employed, with a job but not at work, Armed Forces, with a job but not at work)
  - 1: Car, truck, or van
  - 2: Bus or trolley bus
  - 3: Streetcar or trolley car (carro publico in Puerto Rico)
  - 4: Subway or elevated
  - 5: Railroad
  - 6: Ferryboat
  - 7: Taxicab
  - 8: Motorcycle
  - 9: Bicycle
  - 10: Walked;
  - 11: Worked at home
  - 12: Other method
- POWPUMA (Place of work PUMA based on 2010 Census definitions): Please see ACS PUMS documentation for details on PUMA codes
- POVPIP (Income-to-poverty ratio recode): Range of values:
  - N/A
  - integers 0-500
  - 501 for 501 percent or more

**Filters:**

- AGEP (Age) must be greater than 16.
- PWGTP (Person weight) must be greater than or equal to 1.
- ESR (Employment status recode) must be equal to 1 (employed).

### B.6   Dataset access and license

We provide a flexible software package to download ACS PUMS data and construct both the new prediction tasks discussed in Section 3, as well as new tasks using ACS PUMS data products. The ACS PUMS data itself is governed by the terms of service from the US Census Bureau. For more information, see `https://www.census.gov/data/developers/about/terms-of-service.html` Similarly, the IPUMS adult reconstruction is governed by the IPUMS terms of use. For more information, see `https://ipums.org/about/terms`.

### B.7   Table 1 experiment details

For each of the tasks listed in Table 1 (ACSIncome, ACSPublicCoverage, ACSMobility, ACSEmployment, ACSTravelTime), we use the 1-year 2018 US-Wide ACS PUMS data. We randomly split 80% of the dataset into a training split and the remaining 20% into a test split. All features are standardized to be zero-mean and unit-variance. `Constant Predictor` refers to the majority class baseline, `LogReg` refers to a logistic regression baseline, and `GBM` refers to a gradient boosted decision tree classifier. For each models, we use the implementation provided by Pedregosa et al. [28] with the default hyperparameters.

## C   Tour of empirical observations: missing experimental details

**Models and hyperparameters.**   All of the experiments in this section use the same unconstrained base model: a gradient boosted decision tree (GBM). We chose this model because it trains quickly and

consistently achieved higher accuracy than other baseline models we considered (logistic regression and random forests) in the unconstrained setting; experiments using other base models also produced qualitatively similar results, so we focus on GBM in this paper. We use the implementation provided by Pedregosa et al. [28] and use `exponential` loss, `num_estimators` 5, `max_depth` 5, and all other hyperparameters set to the default. These hyperparameters were chosen via a small grid search to maximize accuracy on the ACSIncome task. We use the implementation of LFR [34] from Bellamy et al. [7] with hyperparameters k=10, `Ax`=0.1, `Ay`=1.0, `Az` = 2.0, `maxiter`=5000, and `maxfun`=5000. The hyperparameters are the same as those used in the UCI Adult tutorial provided by Bellamy et al. [7]. For the in-processing method (ExpGrad) from Agarwal et al. [2], we use the implementation from Bird et al. [9] with the default hyperparameters, and for the post-processing method, we use the threshold adjustment method of Hardt et al. [20], which is also implemented in Bellamy et al. [7]. In Section 4, we use all of the methods to enforce demographic parity. We detail additional experiments enforcing equality of opportunity in Appendix D.

**Datasets.** Throughout this section, we use the ACSIncome task described in Section 3 and Appendix B. With the exception of the distribution shift across time experiments, we use the 2018 1-Year ACS PUMS data. For each state, we randomly split 80% of the dataset into a training split and use the remaining 20% as a test split. The US-Wide dataset is constructed by combining these training and testing sets over all 50 states and Puerto Rico. For the distribution shift across time experiments, we use the same procedure for the 2014-2017 1-Year ACS PUMS data.

**Confidence intervals.** To account for random variation in estimating model accuracies and violations of demographic parity and equality of opportunity, we report each of these metrics with appropriate confidence intervals. We report and plot accuracy numbers with 95% Clopper-Pearson intervals. We report and plot violations of demographic parity and equality of opportunity with 95% Newcombe intervals for the difference between two binomial proportions.

**Compute environment.** All of our experiments are run on CPUs on a cluster computer with 24 Intel Xeon E7 CPUs and 300 GB of RAM.

# D    Additional experiments

In this section, we conduct the same set of experiments conducted in Section 4 on the 5 other prediction tasks we introduced in Section 3. Throughout we keep the experimental details (models, hyperparameters, etc) identical to those detailed in Appendix C.

## D.1    Intervention effect sizes across states

As in Section 4, we train an unconstrained gradient boosted decision tree (GBM) on each state, and we compare the accuracy and fairness criterion violation of this unconstrained model with the same model after applying one of three common fairness intervention: pre-processing (LFR), the in-processing fair reductions methods from Agarwal et al. [2] (ExpGrad), and the simple post-processing method that adjusts group-based acceptance thresholds to satisfy a constraint [20]. Figure 6 shows the result of this experiment for the ACSIncome prediction task for interventions to achieve equality of opportunity.

In Figure 7, we conduct the same experiment for demographic parity on four other ACS data tasks: ACSPublicCoverage, ACSEmployment, ACSMobility, and ACSTravelTime, respectively.

## D.2    Geographic distribution shift

In Figure 8, we plot accuracy and equality of opportunity violation with respect to race for both an unconstrained GBM and the same model after applying a post-processing adjustment to achieve equality of opportunity on a natural suite of test sets: the in-distribution (same state test set) and the out-of-distribution test sets for the 49 other states. This is the same experiment as in Section 4, but with equality of opportunity rather than demographic parity as the metric of interest. In Figures 9, 10 11, and 12 we conduct the same experiment for demographic parity on four other ACS data tasks: ACSPublicCoverage, ACSEmployment, ACSMobility, and ACSTravelTime, respectively.

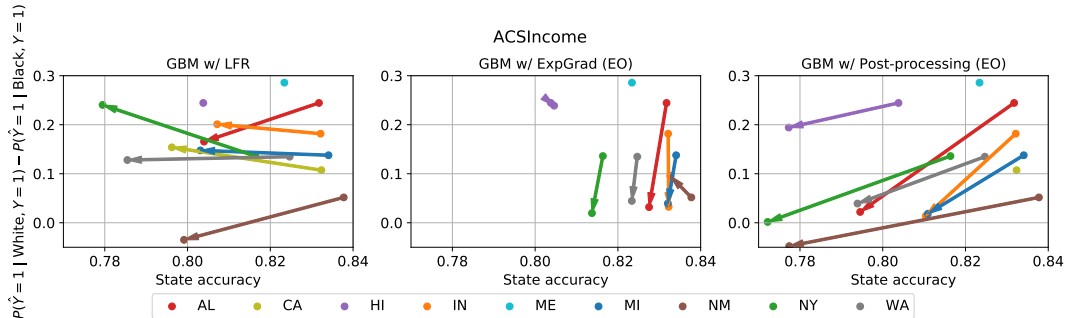

Figure 6: The effect size of fairness interventions varies by state. Each panel shows the change in accuracy and equality of opportunity violation (EO) on the ACSIncome task after applying a fairness intervention to an unconstrained gradient boosted decision tree (GBM). Each arrow corresponds to a different state distribution. The arrow base represents the (accuracy, EO) point corresponding to the unconstrained GBM, and the head represents the (accuracy, EO) point obtained after applying the intervention. The arrow for HI in the LFR plot and ME in all three plots is entirely covered by the start and end points.

## D.3  Temporal distribution shift

In Figure 13, we plot model accuracy and equality of opportunity violation for a GBM trained on the ACSIncome task using US-wide data from 2014 and evaluated on the test sets for the same task drawn from years 2014-2018. This is the same experiment as conducted in Section 4; however, here we consider interventions to satisfy equality of opportunity rather than demographic parity. In Figure 14, we conduct repeat this experiment for interventions to satisfy demographic parity on 4 other ACS PUMS predictions tasks: ACSPublicCoverage, ACSMobility, ACSEmployment, and ACSTravelTime.

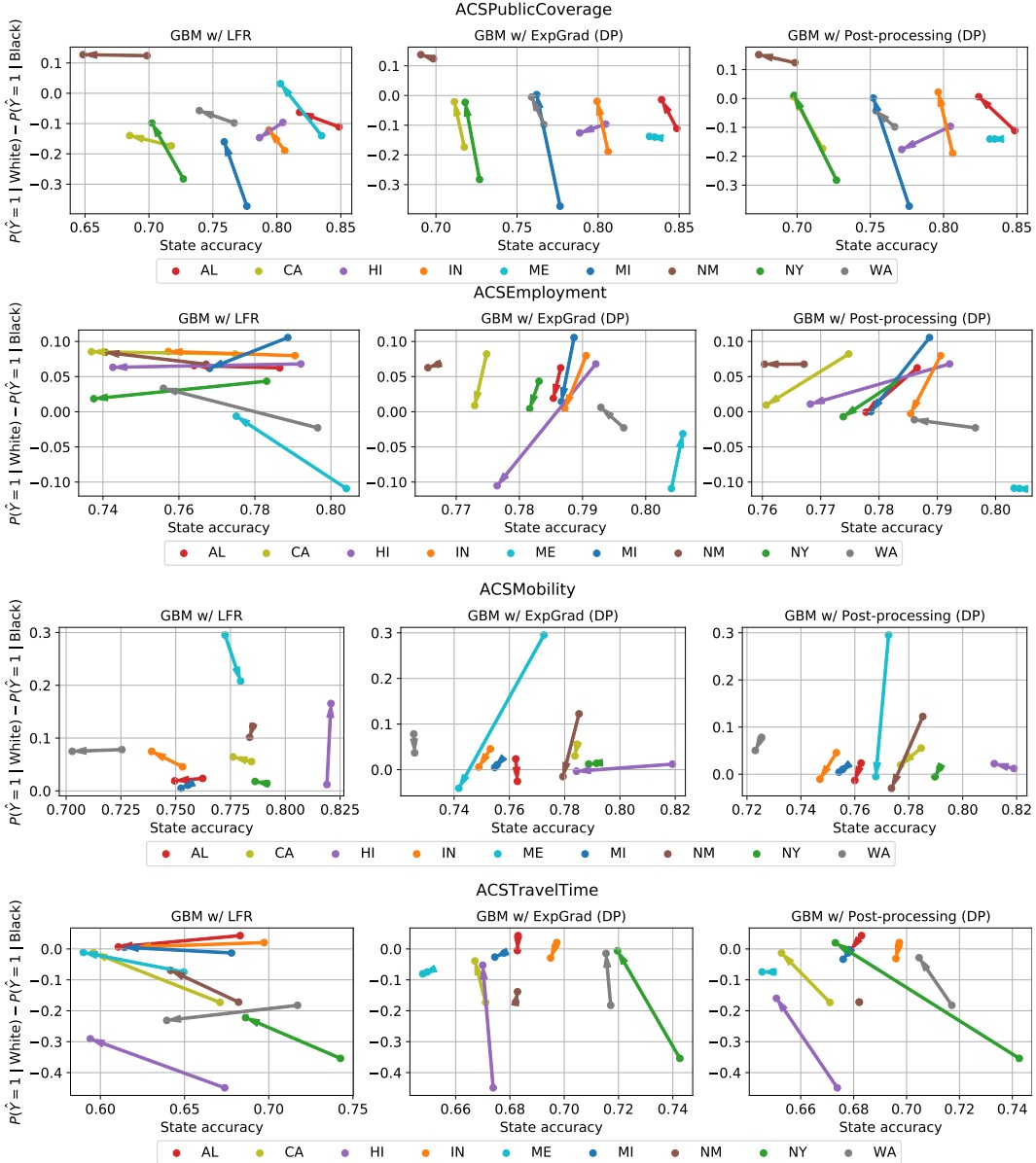

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

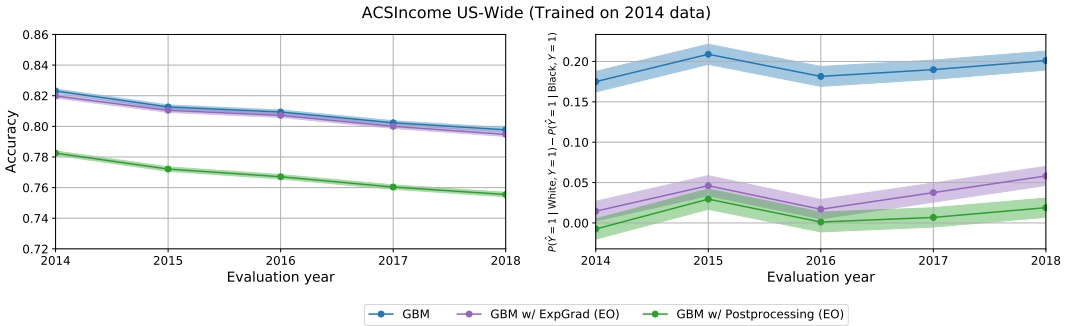

Figure 13: Fairness criteria are more stable over time than accuracy. **Left:** Models trained in 2014 on US-wide ACSIncome with and without fairness interventions to achieve equality of opportunity and evaluated on data in subsequent years. **Right:** Violations of equality of opportunity for the same collection of models. Although accuracy drops over time for most problems, violations of equality of opportunity remain essentially constant. Confidence intervals are 95% Clopper-Pearson intervals for accuracy and 95% Newcombe intervals for equality of opportunity violations.

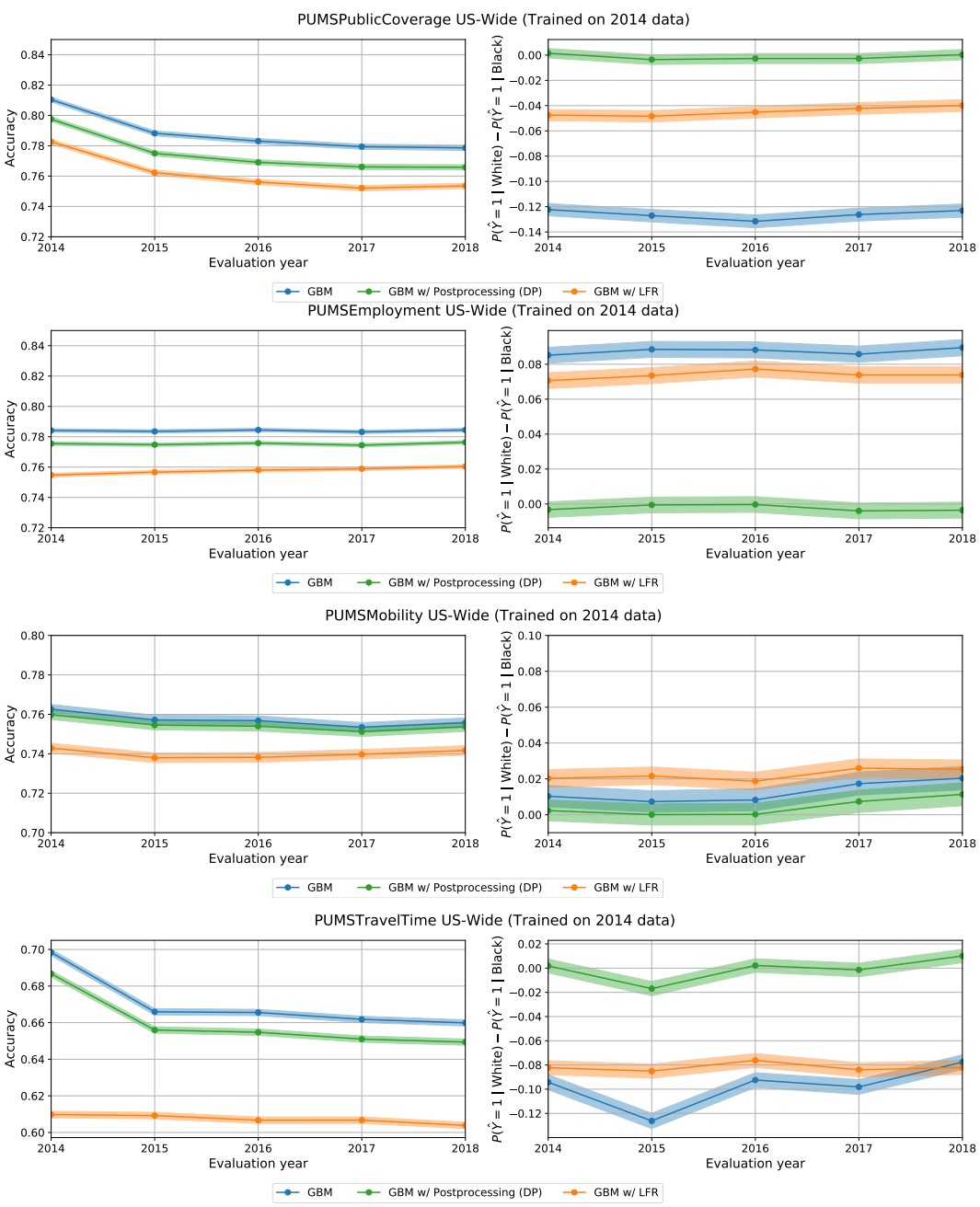

Figure 14: Fairness criteria are more stable over time than accuracy. **Left:** Models trained in 2014 on US-wide ACS data with and without fairness interventions to achieve demographic parity and evaluated on data in subsequent years. **Right:** Violations of demographic parity for the same collection of models. Although accuracy drops over time for most problems, violations of demographic parity remain essentially constant. Confidence intervals are 95% Clopper-Pearson intervals for accuracy and 95% Newcombe intervals for demographic parity.

# E  Datasheet

This datasheet covers both the prediction tasks we introduce and the underlying US Census data sources. However, due to the extensive documentation available about the US Census data we often point to relevant available resources rather than recreating them here. For the most up-to-date version of this datasheet, please refer to `https://github.com/zykls/folktables/blob/main/datasheet.md`.

## E.1  Motivation

- **For what purpose was the dataset created?** Was there a specific task in mind? Was there a specific gap that needed to be filled? Please provide a description.

  The motivation for creating prediction tasks on top of US Census data was to extend the dataset ecosystem available for algorithmic fairness research as outlined in this paper.

- **Who created the dataset (e.g., which team, research group) and on behalf of which entity (e.g., company, institution, organization)?**

  The new prediction tasks were created from available US Census data sources by Frances Ding, Moritz Hardt, John Miller, and Ludwig Schmidt.

- **Who funded the creation of the dataset?** If there is an associated grant, please provide the name of the grantor and the grant name and number.

  Frances Ding, Moritz Hardt, and John Miller were employed by the University of California for the duration of this research project, funded by grants administered through the University of California. Ludwig Schmidt was employed by Toyota Research throughout this research project.

- **Any other comments?**

  No.

## E.2  Composition

- **What do the instances that comprise the dataset represent (e.g., documents, photos, people, countries)?** Are there multiple types of instances (e.g., movies, users, and ratings; people and interactions between them; nodes and edges)? Please provide a description.

  Each instance in our IPUMS Adult reconstruction represents an individual. Similarly, our datasets derived from ACS contains instances representing individuals. The ACS data our datasets are derived from also contain household-level information and the relationship between households and individuals.

- **How many instances are there in total (of each type, if appropriate)?**

  Our IPUMS Adult reconstruction contains 49,531 rows (see Section 2.1). Table 1 contains the sizes of our datasets derived from ACS.

- **Does the dataset contain all possible instances or is it a sample (not necessarily random) of instances from a larger set?** If the dataset is a sample, then what is the larger set? Is the sample representative of the larger set (e.g., geographic coverage)? If so, please describe how this representativeness was validated/verified. If it is not representative of the larger set, please describe why not (e.g., to cover a more diverse range of instances, because instances were withheld or unavailable)

  Both IPUMS Adult and our ACS datasets are samples of the US population. Please see Sections 2.1 & 3 and the corresponding documentation provided by the US Census Bureau. Note that the per-instance weights have to be taken into account if the sample is meant to represent the US population.

- **What data does each instance consist of?** "Raw" data (e.g., unprocessed text or images) or features? In either case, please provide a description.

  Each instance consists of features. IPUMS Adult uses the same features as the original UCI Adult dataset. Appendix B describes each feature in our new datasets derived from ACS.

- **Is there a label or target associated with each instance?** If so, please provide a description.

  Similar to UCI Adult, our IPUMS Adult reconstruction uses the income as label (where the continuous values as opposed to only the binarized values are now available). Appendix B describes the labels in our new datasets derived from ACS.

- **Is any information missing from individual instances?** If so, please provide a description, explaining why this information is missing (e.g., because it was unavailable). This does not include intentionally removed information, but might include, e.g., redacted text.

  Some features (e.g., the country of origin in IPUMS Adult) contain missing values. We again refer to the respective documentation from the US Census Bureau for details.

- **Are relationships between individual instances made explicit (e.g., users' movie ratings, social network links)?** If so, please describe how these relationships are made explicit.

  Our versions of the datasets contain no relationships between individuals. The original data sources from the US Census contain relationships between individuals and households.

- **Are there recommended data splits (e.g., training, development/validation, testing)?** If so, please provide a description of these splits, explaining the rationale behind them.

  For IPUMS Adult, it is possible to follow the same train / test split as the original UCI Adult. In general, we recommend k-fold cross-validation for all of our datasets.

- **Are there any errors, sources of noise, or redundancies in the dataset?** If so, please provide a description.

  Our IPUMS Adult reconstruction contains slightly more rows than the original UCI Adult, see Section 2.1. Beyond IPUMS Adult, we refer to the documentation of CPS and ACS provided by the US Census Bureau.

- **Is the dataset self-contained, or does it link to or otherwise rely on external resources (e.g., websites, tweets, other datasets)?** If it links to or relies on external resources, a) are there guarantees that they will exist, and remain constant, over time; b) are there official archival versions of the complete dataset (i.e., including the external resources as they existed at the time the dataset was created); c) are there any restrictions (e.g., licenses, fees) associated with any of the external resources that might apply to a future user? Please provide descriptions of all external resources and any restrictions associated with them, as well as links or other access points, as appropriate.

  Due to restrictions on the re-distribution of the original IPUMS and ACS data sources, we do not provide our datasets as standalone data files. Instead, we provide scripts to generate our datasets from the respective sources.

  Both the US Census Bureau and IPUMS aim to provide stable long-term access to their data. Hence we consider these data sources to be reliable. We refer to the IPUMS website and the website of the US Census Bureau for specific usage restrictions. Neither data source has fees associated with it.

- **Does the dataset contain data that might be considered confidential (e.g., data that is protected by legal privilege or by doctor patient confidentiality, data that includes the content of individuals' non-public communications)?** If so, please provide a description.

  Our datasets are subsets of datasets released publicly by the US Census Bureau.

- **Does the dataset contain data that, if viewed directly, might be offensive, insulting, threatening, or might otherwise cause anxiety?** If so, please describe why.

  No.

- **Does the dataset relate to people?** If not, you may skip the remaining questions in this section.

  Yes, each instance in our datasets corresponds to a person.

- **Does the dataset identify any subpopulations (e.g., by age, gender)?** If so, please describe how these subpopulations are identified and provide a description of their respective distributions within the dataset.

  Our datasets identify subpopulations since each individual has features such as age, gender, or race. Please see the main text of our paper for experiments exploring the respective distributions.

- **Is it possible to identify individuals (i.e., one or more natural persons), either directly or indirectly (i.e., in combination with other data) from the dataset?** If so, please describe how.

  To the best of our knowledge, it is not possible to identify individuals *directly* from our datasets. However, the possibility of reconstruction attacks combining data from the US Cenus Bureau (such as CPS and ACS) and other data sources are a concern and actively investigated by the research community.

- **Does the dataset contain data that might be considered sensitive in any way (e.g., data that reveals racial or ethnic origins, sexual orientations, religious beliefs, political opinions or union memberships, or locations; financial or health data; biometric or genetic data; forms**

**of government identification, such as social security numbers; criminal history)?** If so, please provide a description.

Our datasets contain features such as race, age, or gender that are often considered sensitive. This is by design since we assembled our datasets to test algorithmic fairness interventions.

- **Any other comments?**

No.

### E.3 Collection process

- **How was the data associated with each instance acquired?** Was the data directly observable (e.g., raw text, movie ratings), reported by subjects (e.g., survey responses), or indirectly inferred/derived from other data (e.g., part-of-speech tags, model-based guesses for age or language)? If data was reported by subjects or indirectly inferred/derived from other data, was the data validated/verified? If so, please describe how.

  The data was reported by subjects as part of the ACS and CPS surveys. The respective documentation provided by the US Census Bureau contains further information, see `https://www.census.gov/programs-surveys/acs/methodology/design-and-methodology.html` and `https://www.census.gov/programs-surveys/cps/technical-documentation/methodology.html`.

- **What mechanisms or procedures were used to collect the data (e.g., hardware apparatus or sensor, manual human curation, software program, software API)?** How were these mechanisms or procedures validated?

  The ACS relies on a combination of internet, mail, telephone, and in-person interviews. CPS uses in-person and telephone interviews. Please see the aforementioned documentation from the US Census Bureau for detailed information.

- **If the dataset is a sample from a larger set, what was the sampling strategy (e.g., deterministic, probabilistic with specific sampling probabilities)?**

  For the ACS, the US Census Bureau sampled housing units uniformly for each county. See Chapter 4 in the ACS documentation (`https://www2.census.gov/programs-surveys/acs/methodology/design_and_methodology/acs_design_methodology_report_2014.pdf`) for details.

  CPS is also sampled by housing unit from certain sampling areas, see Chapters 3 and 4 in `https://www.census.gov/prod/2006pubs/tp-66.pdf`.

- **Who was involved in the data collection process (e.g., students, crowdworkers, contractors) and how were they compensated (e.g., how much were crowdworkers paid)?**

  The US Census Bureau employs interviewers for conducting surveys. According to online job information platforms such as `indeed.com`, an interviewer earns about $15 per hour.

- **Over what timeframe was the data collected?** Does this timeframe match the creation timeframe of the data associated with the instances (e.g., recent crawl of old news articles)? If not, please describe the timeframe in which the data associated with the instances was created.

  Both CPS and ACS collect data annually. Our IPUMS Adult reconstruction contains data from the 1994 CPS ASEC. Our new tasks derived from ACS can be instantiated for various survey years.

- **Were any ethical review processes conducted (e.g., by an institutional review board)?** If so, please provide a description of these review processes, including the outcomes, as well as a link or other access point to any supporting documentation.

  Both ACS and CPS are regularly reviewed by the US Census Bureau. As a government agency, the US Census Bureau is also subject to government oversight mechanisms.

- **Does the dataset relate to people? If not, you may skip the remainder of the questions in this section.**

  Yes.

- **Did you collect the data from the individuals in question directly, or obtain it via third parties or other sources (e.g., websites)?**

  Data collection was performed by the US Census Bureau. We obtained the data from publicly available US Census repositories.

- **Were the individuals in question notified about the data collection?** If so, please describe (or show with screenshots or other information) how notice was provided, and provide a link or other access point to, or otherwise reproduce, the exact language of the notification itself.

  Yes. A sample ACS form is available online: `https://www.census.gov/programs-surveys/acs/about/forms-and-instructions/2021-form.html`

  Information about the CPS collection methodology is available here: `https://www.census.gov/programs-surveys/cps/technical-documentation/methodology.html`

- **Did the individuals in question consent to the collection and use of their data?** If so, please describe (or show with screenshots or other information) how consent was requested and provided, and provide a link or other access point to, or otherwise reproduce, the exact language to which the individuals consented.

  Participation in the US Census American Community Survey is mandatory. Participation in the US Corrent Population Survey is voluntary and consent is obtained at the beginning of the interview: `https://www2.census.gov/programs-surveys/cps/methodology/CPS-Tech-Paper-77.pdf`

- **If consent was obtained, were the consenting individuals provided with a mechanism to revoke their consent in the future or for certain uses?** If so, please provide a description, as well as a link or other access point to the mechanism (if appropriate).

  We are not aware that the Census Bureau would provide such a mechanism.

- **Has an analysis of the potential impact of the dataset and its use on data subjects (e.g., a data protection impact analysis) been conducted?** If so, please provide a description of this analysis, including the outcomes, as well as a link or other access point to any supporting documentation.

  The US Census Bureau assesses privacy risks and invests in statistical disclosure control. See `https://www.census.gov/topics/research/disclosure-avoidance.html`. Our derived prediction tasks do not increase privacy risks.

- **Any other comments?**

  No.

### E.4 Preprocessing / cleaning / labeling

- **Was any preprocessing/cleaning/labeling of the data done (e.g., discretization or bucketing, tokenization, part-of-speech tagging, SIFT feature extraction, removal of instances, processing of missing values)?** If so, please provide a description. If not, you may skip the remainder of the questions in this section.

  We used two US Census data products – we reconstructed UCI Adult from the Annual Social and Economic Supplement (ASEC) of the Current Population Survey (CPS), and we constructed new prediction tasks from the American Community Survey (ACS) Public Use Microdata Sample (PUMS). Before releasing CPS data publicly, the Census Bureau top-codes certain variables and conducts imputation of certain missing values, as documented here: `https://www.census.gov/programs-surveys/cps/technical-documentation/methodology.html`. In our IPUMS Adult reconstruction, we include a subset of the variables available from the CPS data and do not alter their values.

  The ACS data release similarly top-codes certain variables and conducts imputation of certain missing values, as documented here: `https://www.census.gov/programs-surveys/acs/microdata/documentation.html`. For the new prediction tasks that we define, we further process the ACS data as documented at the folktables GitHub page, `https://github.com/zykls/folktables`. In most cases, this involves mapping missing values (NaNs) to $-1$. We release code so that new prediction tasks may be defined on the ACS data, with potentially different preprocessing. Each prediction task also defines a binary label by discretizing the target variable into two classes; this can be easily changed to define a new labeling in a new prediction task.

- **Was the "raw" data saved in addition to the preprocessed/cleaned/labeled data (e.g., to support unanticipated future uses)?** If so, please provide a link or other access point to the "raw" data.

  Yes, our package provides access to the data as released by the U.S. Census Bureau. The "raw" survey answers collected by the Census Bureau are not available for public release due to privacy considerations.

- **Is the software used to preprocess/clean/label the instances available?** If so, please provide a link or other access point.

  The software to is available at the folktables GitHub page, `https://github.com/zykls/folktables`.

- **Any other comments?**

  No.

## E.5 Uses

- **Has the dataset been used for any tasks already?** If so, please provide a description.

  In this paper we create five new prediction tasks from the ACS PUMS data:

  1. ACSIncome: Predict whether US working adults' yearly income is above $50,000.
  2. ACSPublicCoverage: Predict whether a low-income individual, not eligible for Medicare, has coverage from public health insurance.
  3. ACSMobility: Predict whether a young adult moved addresses in the last year.
  4. ACSEmployment: Predict whether a US adult is employed.
  5. ACSTravelTime: Predict whether a working adult has a travel time to work of greater than 20 minutes.

  Further details about these tasks can be found at the folktables GitHub page, `https://github.com/zykls/folktables`, and in Appendix B.

- **Is there a repository that links to any or all papers or systems that use the dataset?** If so, please provide a link or other access point.

  At the folktables GitHub page, `https://github.com/zykls/folktables`, any public forks to the package are visible, and papers or systems that use the datasets should cite the paper linked at that Github page.

- **What (other) tasks could the dataset be used for?**

  New prediction tasks may be defined on the ACS PUMS data that use different subsets of variables as features and/or different target variables. Different prediction tasks may have different properties such as Bayes error rate, or the base rate disparities between subgroups, that can help to benchmark machine learning models in diverse settings.

- **Is there anything about the composition of the dataset or the way it was collected and preprocessed/cleaned/labeled that might impact future uses?** For example, is there anything that a future user might need to know to avoid uses that could result in unfair treatment of individuals or groups (e.g., stereotyping, quality of service issues) or other undesirable harms (e.g., financial harms, legal risks) If so, please provide a description. Is there anything a future user could do to mitigate these undesirable harms?

  Both the CPS and ACS are collected through surveys of a subset of the US population, and in their documentation, they acknowledge that statistical trends in individual states may be noisy compared to those found by analyzing US data as a whole, due to small sample sizes in certain states. In particular, there may be very few individuals with particular characteristics (e.g. ethnicity) in certain states, and generalizing conclusions from these few individuals may be highly inaccurate. Further, benchmarking fair machine learning algorithms on datasets with few representatives of certain subgroups may provide the illusion of "checking a box" for fairness, without substantive merit.

- **Are there tasks for which the dataset should not be used?** If so, please provide a description.

  This dataset contains personal information, and users should not attempt to re-identify individuals in it. Further, these datasets are meant primarily to aid in benchmarking machine learning algorithms; Census data is often crucial for substantive, domain-specific work by social scientists, but our dataset contributions are not in this direction. Substantive investigations into inequality, demographic shifts, and other important questions should not be based purely on the datasets we provide.

- **Any other comments?**

  No.

### E.6 Distribution

- **Will the dataset be distributed to third parties outside of the entity (e.g., company, institution, organization) on behalf of which the dataset was created?** If so, please provide a description.

  The dataset will be available for public download on the folktables GitHub page, `https://github.com/zykls/folktables`.

- **How will the dataset will be distributed (e.g., tarball on website, API, GitHub)?** Does the dataset have a digital object identifier (DOI)?

  The dataset will be be distributed via GitHub, see `https://github.com/zykls/folktables`. The dataset does not have a DOI.

- **When will the dataset be distributed?**

  The dataset will be released on August 1, 2021 and available thereafter for download and public use.

- **Will the dataset be distributed under a copyright or other intellectual property (IP) license, and/or under applicable terms of use (ToU)?** If so, please describe this license and/or ToU, and provide a link or other access point to, or otherwise reproduce, any relevant licensing terms or ToU, as well as any fees associated with these restrictions.

  The folktables package and data loading code will be available under the MIT license. The folktables data itself is based on data from the American Community Survey (ACS) Public Use Microdata Sample (PUMS) files managed by the US Census Bureau, and it is governed by the terms of use provided by the Census Bureau. For more information, see `https://www.census.gov/data/developers/about/terms-of-service.html`

  Similarly, the IPUMS adult reconstruction is governed by the IPUMS terms of use. For more information, see `https://ipums.org/about/terms`.

- **Have any third parties imposed IP-based or other restrictions on the data associated with the instances?** If so, please describe these restrictions, and provide a link or other access point to, or otherwise reproduce, any relevant licensing terms, as well as any fees associated with these restrictions.

  The folktables data and the adult reconstruction data are governed by third-party terms of use provided by the US Census Bureau and IPUMS, respectively. See `https://www.census.gov/data/developers/about/terms-of-service.html` and `https://ipums.org/about/terms` for complete details. The IPUMS Adult Reconstruction is a subsample of the IPUMS CPS data available from `cps.ipums.org` These data are intended for replication purposes only. Individuals analyzing the data for other purposes must submit a separate data extract request directly via IPUMS CPS. Individuals should contact `ipums@umn.edu` for redistribution requests.

- **Do any export controls or other regulatory restrictions apply to the dataset or to individual instances?** If so, please describe these restrictions, and provide a link or other access point to, or otherwise reproduce, any supporting documentation.

  To our knowledge, no export controls or regulatory restrictions apply to the dataset.

- **Any other comments?**
  No.

### E.7 Maintenance

- **Who is supporting/hosting/maintaining the dataset?**

  The dataset will be hosted on GitHub, and supported and maintained by the folktables team. As of June 2021, this team consists of Frances Ding, Moritz Hardt, John Miller, and Ludwig Schmidt.

- **How can the owner/curator/manager of the dataset be contacted (e.g., email address)?**

  Please send issues and requests to `folktables@gmail.com`.

- **Is there an erratum?** If so, please provide a link or other access point.

  An erratum will be hosted on the dataset website, `https://github.com/zykls/folktables`.

- **Will the dataset be updated (e.g., to correct labeling errors, add new instances, delete instances)?** If so, please describe how often, by whom, and how updates will be communicated to users (e.g., mailing list, GitHub)?

The dataset will be updated as required to address errors and refine the prediction problems based on feedback from the community. The package maintainers will update the dataset and communicate these updates on GitHub.

- **If the dataset relates to people, are there applicable limits on the retention of the data associated with the instances (e.g., were individuals in question told that their data would be retained for a fixed period of time and then deleted)?** If so, please describe these limits and explain how they will be enforced.

  The data used in folktables is based on data from the American Community Survey (ACS) Public Use Microdata Sample (PUMS) files managed by the US Census Bureau. The data inherits and will respect the corresponding retention policies of the ACS. Please see `https://www.census.gov/programs-surveys/acs/about.html` for more details. For the Adult reconstruction dataset, the data is based on Current Population Survey (CPS) released by IPUMS and thus inherits and will respect the corresponding retention policies for the CPS. Please see `https://cps.ipums.org/cps/` for more details.

- **Will older versions of the dataset continue to be supported/hosted/maintained?** If so, please describe how. If not, please describe how its obsolescence will be communicated to users.

  Older versions of the datasets in folktables will be clearly indicated, supported, and maintained on the GitHub website. Each new version of the dataset will be tagged with `version` metadata and an associated GitHub release.

- **If others want to extend/augment/build on/contribute to the dataset, is there a mechanism for them to do so?** If so, please provide a description. Will these contributions be validated/verified? If so, please describe how. If not, why not? Is there a process for communicating/distributing these contributions to other users? If so, please provide a description.

  Users wishing to contribute to folktables datasets are encouraged to do so by submitting a pull request on the website `https://github.com/zykls/folktables/pulls`. The contributions will be reviewed by the maintainers. These contributions will be reflected in new version of the dataset and broadcasted as part of each Github release.

- **Any other comments?**

  No.