# OpenReview forum: "Retiring Adult: New Datasets for Fair Machine Learning"
_NeurIPS.cc/2021/Conference — NeurIPS 2021 Oral_

### Official Review · Reviewer_vtAB · 2021-07-06

**Rating:** 7
**Confidence:** 4

**Summary:**

They have reconstructed the famous UCI adult dataset to seemingly high accuracy, using the CPS data that provides granularity at the individual level. Furthermore, as their reconstructed dataset now spans multiple years and all states of the United States as opposed to the original adult dataset which I believe is specific to 1994, they were able to created other related datasets for different prediction tasks.

They have carried out a bunch of experiments that demonstrate how different seemingly innocuous decisions can lead to very different outcomes. For example, because they have the raw data for the income as opposed to the original adult dataset which indicates whether the income was above 50k or not, they can see how varying the income threshold can lead to different outcomes.

**Limitations And Societal Impact:**

Is there any concern for privacy? For instance, is it possible that one can use the methodology used here to reconstruct the adult dataset to reconstruct dataset sets for more recent years? Or does CPS data not available for recent years for privacy reasons?

**Main Review:**

Originality:

I don't know of any prior work trying to reconstruct the adult dataset and try to understand some of the contexts behind the dataset. Also, the new datasets (as a result of being able to span multiple years and states) are novel.

Quality:

Their discussion of the reconstruction methodology (e.g. section 2.1) is convincing in that I believe that they have reconstructed the dataset with very high accuracy.

Clarity:

Overall, the paper reads very easily in terms of its presentation of the results.

Strength:

Having the actual raw income data from which the adult data's feature (>50k, <= 50k) is derived is important. By choosing a threshold at different quantiles, it allows other researchers to induce different distributions and understand the effect of different algorithms in those different distributions. I think this point can be even more emphasized in the paper.

Weakness:

A lot of the conclusions derived from the experiments seem to all point to the fact that each state's distribution is very different from one another whereas there isn't that much distribution shift in time.

The tradeoff between accuracy and fairness heavily depends on the distribution itself, so it is intuitive that different states see different effectiveness in intervention. If there's a large discrepancy between the training and test distribution, one cannot expect the classifier trained on the training dataset to behave nicely on the test distribution.

Also, the experiment purporting to show that increased datasest doesn't mitigate observed disparities is a little weird in that it is looking at two different distributions (1994 vs 2018) of different sizes as opposed to the same distribution of different sizes (same year but one is subsampled from the other).

So, I think it may be more helpful to emphasize that these datasets allow one to see that there is indeed much variation in distributions across states and in some other dimensions — and the consequences of such variation — instead of making separate claims about the nature of the dataset.

Questions/comments:

-The description of CPS doesn't show up until section 3 (e.g. that it stands for Current Population Survey), but this acronym is used without much discussion already in section 2.1. Similarly, what does IPUMS stand for?

-------------------------------------------------------------------------------------------------------------------------------------------------------------------------------------------
Thanks for answering the questions! Despite my hesitancy about technical findings of the new dataset, I agree that the new dataset alone is a valuable contribution. I will be bumping up the score to 7

**Time Spent Reviewing:**

3

---

> ### Author Response · Authors · 2021-08-10
> **Author response**
>
> We thank the reviewer for their comments and feedback.
>
> The reviewer’s primary concern is that many of the empirical results in the paper stem from the observation that states have very different distributions, whereas the distribution shift in time is weaker. However, this finding is itself one of the primary empirical contributions of our work, and it was not a priori obvious without running the experiments in the paper. This observation is important in itself, and, as the reviewer notes, suggests caution when using results from one state to justify a particular fairness intervention in another.
>
> Moreover, the differences in intervention effectiveness across states hints at the diversity of distributions and test environments that our datasets offer for analyzing and evaluating new methods. This is particularly beneficial in algorithmic fairness, which previously relied on a very limited number of distributions (primarily UCI Adult) to benchmark and study algorithm performance.
>
> Regarding the “experiment purporting to show that increased dataset size doesn't mitigate observed disparities,” the aim of the experiment is to show both that (a) passage of time from 1994-2018 and (b) the larger dataset size does not necessarily mitigate observed disparities, despite conventional wisdom. We will make this more clear in the revision. Furthermore, to isolate the effects of the passage of time and dataset size, we will also report results with the original and a subsampled version of the 2018 data.
>
> Regarding privacy risk, all of our datasets make use of publicly available data from the US Census Bureau. The US Census Bureau assesses privacy risks and also invests in disclosure avoidance and statistical disclosure control. See https://www.census.gov/topics/research/disclosure-avoidance.html for more details.  Our derived prediction tasks do not increase privacy risks to individuals.
>
> Minor:
> In the revised version, we will more carefully define the various acronyms we use, (e.g. CPS -> Current Population Survey). IPUMS refers to the Integrated Public Use Microdata Series and is the broader organization which tracks and publishes the historical CPS data we use (under the name IPUMS CPS).

---

### Official Review · Reviewer_7q2x · 2021-07-08

**Rating:** 8
**Confidence:** 4

**Summary:**

This paper provides a new dataset on which to ask questions around fairness in machine learning, intended as a replacement for the frequently-used UCI Adult dataset. The authors attempt to reconstruct the Adult dataset from US Census data, provide a broader dataset based on the American Community Survey Public Use Microdata Sample with several predefined prediction tasks, and perform an initial analysis of the new dataset from a fairness perspective.

**Limitations And Societal Impact:**

The authors make certain choices in constructing the datasets, in particular with respect to the prediction tasks they define. It may be worth elaborating on those choices from the perspective of what types of outcomes they're encouraging others to predict.

**Main Review:**

This paper makes a solid contribution to the field of fairness in machine learning. The authors show how it can be used to ask a variety of new questions around temporal and geographical differences, among others.

One minor point that I feel is worth including -- the ACS data includes some noise due to the Census Disclosure Avoidance System, and it's not fully known how this operated in 1994 (see, e.g., "A History of the Current Population Survey and Disclosure Avoidance" (McKenna, 2019)).

Other than that, I don't have major comments -- the paper is well-written and makes an important contribution.

**Time Spent Reviewing:**

2

---

> ### Author Response · Authors · 2021-08-10
> **Author response**
>
> We thank the reviewer for their comments and feedback. In the final version of the paper, we will include a note on the Census Disclosure Avoidance System. To avoid confusion, we used the 1994 CPS data, not the 1994 ACS data to reconstruct UCI Adult. Further, we use the 1994 CPS data only for the Adult reconstruction, and we are able to completely reconstruct Adult (up to the weighting column ‘fnlwgt’) without knowledge of the DA system in 1994. We will also include some additional discussion around the choice of predefined predictions tasks.

---

> > ### Comment · Reviewer_7q2x · 2021-09-03
> > **Thanks for your response**
> >
> > Having read the other reviews and responses, I'm keeping my score as is.

---

### Official Review · Reviewer_hnqZ · 2021-07-20

**Rating:** 8
**Confidence:** 4

**Summary:**

The authors present a suite of datasets derived from recent US Census survey data and covering a variety of prediction tasks based on tabular demographic data. One of their datasets is a superset of the popular UCI Adult dataset, which is commonly used in many algorithmic fairness papers. The authors show the effect of various dataset design choices (such as the threshold used to binarize income) on accuracy and fairness metrics, as well as on the conclusions one might draw about different fairness interventions. The authors also provide a preliminary analysis of the other datasets, showcasing vignettes such as the effect of distribution shifts over states and time, and the effect of increasing dataset size on fairness metrics.

I thought this was an excellent, well-written paper that addresses an important need within the algorithmic fairness community, namely, the complete lack of high-quality datasets. Beyond the dataset contribution, the empirical analysis that the authors provide is interesting and highlights the potential of these datasets. Kudos to the authors for this thorough and careful work.

------

Update: Thanks to the authors for their detailed responses. I would like to keep my score: I think this is a strong and positive contribution.

**Limitations And Societal Impact:**

Overall, the authors have done a commendable job addressing the limitations and societal impact of their paper. My main feedback in this direction is that the authors try to divorce the application from the ML task, in the sense that it almost doesn't matter for the authors' purposes what the actual prediction task is or where the data comes from. However, I think this perspective has been a persistent limitation of the work on algorithmic fairness, in that it deals with an abstract notion of "fairness" that can be hard to translate into a real application. An ideal benchmark in algorithmic fairness would not only be high-quality from an ML point of view (lots of data, clean, etc.) but also compelling as an actual application.

**Main Review:**

As mentioned above, I think this is a strong paper that will be a positive contribution to the field of algorithmic fairness, and more generally, to the development of ML methods for tabular data. (In hindsight, it is an unfortunate reflection of the state of the field of algorithmic fairness that it took this long to generate better datasets from the publicly-available US Census data.)

I'll focus here on some suggestions and questions:

1. Why are the 5 prediction tasks outlined in the paper useful? Is the argument that there are real-world settings where we would, for example, want to predict if an individual's income is above a threshold given other demographic information, or predict if an individual is covered by public health insurance? Given that the empirical phenomena (the behavior of fairness interventions, the effect of distribution shifts, etc.) can vary significantly from task to task, it would be helpful to understand why we should be keyed to these tasks in particular.

2. If, as claimed, the models are already close to Bayes optimal on these datasets, then what is the point of these datasets from the perspective of ML research? It would be interesting to hear the authors' thoughts on this. One perspective, for example, is that given the impossibility results around finding the one true perfect classifier, if our models are already obtaining Bayes optimal error rates on these datasets, then it is a solved problem as far as ML research goes (in the sense that the right tradeoffs between various notions of accuracy and fairness are questions for society and policymakers to decide, as opposed to algorithmic ones). Should the fairness community focus on the "cognitive machine learning" tasks that the authors note?

3. What is driving the difference between in-distribution and out-of-distribution accuracy? To what extent can this be solved, e.g., if an (oracle) model is trained on all or multiple states, does it close the gap? The discussion from the authors about how the models are close to Bayes optimal seems to suggest that the differences in performance between states might be unavoidable, in the sense that they simply reflect the properties of different data distributions in each state (for example, perhaps some states have data that's noisier; or perhaps label balance varies, so it always helps to learn the right bias for the in-distribution state).

Nit: typo in L1194, "oover"

**Time Spent Reviewing:**

3

---

> ### Author Response · Authors · 2021-08-10
> **Author response**
>
> We thank the reviewer for their comments and feedback. Below we discuss some of the questions the reviewer raised:
>
> “Why are the prediction tasks outlined in the paper useful?” Are there real-world settings that map onto the existing prediction tasks?
>
> This is indeed a subtle question, and one that we attempt to address with our “scope and limitations” discussion in Section 3.2. Our prediction tasks use Census data to address empirical questions within algorithmic fairness, e.g., the comparison of methods for achieving a given fairness constraint, and not to answer substantive questions around inequality in income, education, healthcare, etc. At this stage, we view the diversity of empirical phenomena between tasks as a benefit which can increase the comprehensiveness of empirical evaluations in future work on this topic. Better understanding which tasks and phenomena are most relevant for a given application is an excellent starting point for future work. We will expand on this point and add more discussion to the final version of the paper.
>
> Furthermore, the tasks outlined in the paper are intended to be a starting point for further inquiry. The software released with the paper and datasets includes an easy-to-use API that allows users to construct new prediction tasks with ACS Census data, and we encourage the community to explore additional prediction tasks beyond those we initially defined.
>
> “If the models are close to Bayes optimal, what is the point of these datasets from the perspective of ML research?”
> At the outset, it’s worth noting that the closeness to Bayes optimality is a conjecture, and it is a great follow-up project to rigorously understand whether or not this is true.
>
> If this conjecture holds, the prediction tasks and census data we expose also may be useful for studying tabular datasets more broadly. Using our datasets and dataset construction pipeline, we could generate many tabular datasets varying the population, feature sets, prediction target, and time period and study to what extent these diverse datasets show similar phenomena. Do GBMs always achieve close to the Bayes error, and if not, what properties of the dataset are important for this to hold?
>
> Should the fairness community focus on the "cognitive machine learning" tasks that the authors note?
>
> While much of the discourse in the fairness community has focused on cognitive machine learning tasks, many of the highest profile fairness settings involve tabular data. This includes recidivism prediction [1], failure to appear prediction [2], grade standardization [3], healthcare costs [4], and credit scoring [5]. This list is far from exhaustive and includes examples from domains spanning policing, health, education, and finance. Consequently, we believe tabular datasets are an important focus area, and it is important to understand the empirical phenomena that arise in the tabular setting.
>
> “What is driving the difference between in-distribution and out-of-distribution accuracy?”
>
> This is a very good question and not one we have a complete answer to at the moment. As the reviewer suggested, even for an “oracle” model trained on all of the states, there is still some variation between its performance on each of the states. This hints that some difference may be unavoidable, and understanding the sources of these differences is an excellent direction for further work with our datasets.
>
> [1] Angwin, Julia, et al. "Machine bias: There’s software used across the country to predict future criminals. And it’s biased against blacks.” ProPublica (2016).
>
> [2] Eubanks, Virginia. Automating inequality: How high-tech tools profile, police, and punish the poor. St. Martin's Press, 2018.
>
> [3] Ferguson, Donna; Savage, Michael (15 August 2020). "Controversial exams algorithm to set 97% of GCSE results". The Guardian.
>
> [4] Obermeyer, Ziad, et al. "Dissecting racial bias in an algorithm used to manage the health of populations." Science 366.6464 (2019): 447-453.
>
> [5] Pasquale, Frank. The black box society. Harvard University Press, 2015.

---

> > ### Comment · Reviewer_hnqZ · 2021-08-29
> > **Thanks**
> >
> > Thank you for the detailed response and for your plans to include these additional points in the discussion!

---

### Official Review · Reviewer_raqf · 2021-07-21

**Rating:** 7
**Confidence:** 5

**Summary:**

The authors focus on the adult dataset, which was originated from the US census data and has records of individuals with a binary label of whether the income is higher or lower than 50k. The adult dataset has been one of the gold standard datasets in the algorithmic fair ML community, especially to test for the performance of new fairness metrics and fair algorithms. The authors discuss few drawbacks of the adult dataset and reconstructed a superset of the UCI adult dataset from the available census datasets released in the same year, namely the Annual Social and Economic Supplement (ASEC). They refer to their reconstruction as IPUMS adult. Furthermore, the authors also looked into the American Community Survey (ACS) Public Use Microdata Sample (PUMS) and defined multiple prediction tasks based on the ACS PUMS dataset.

-------------------------------------------------------------------------------------------------------------------------

update: Thanks for authors' comments. I'll be increasing my score as I think this is an important contribution to the community, although I still think that the level of technical novelty is below what is expected of NeurIPS main track.

**Limitations And Societal Impact:**

Discussed in the main review.

**Main Review:**

Like many other ML researchers, I worked with the adult dataset a few times. I also always wished that I had access not only to the binary labels for the income being higher/lower than 50k but also to the actual income values. This is a great contribution, looking more closely into the adult dataset, discussing how different fairness criteria and fair algorithms differ in their fairness and accuracy based on the threshold, discussing more generally how state-wide analysis could be different from the country-wide analysis and many other aspects. I’m just not sure, that given the technical contribution of the paper if the scope of technical novelty would fit what has been traditionally the acceptance bar for NeurIPS and whether NeurIPS is the correct venue for this work. But I’d like to see it published somewhere, and this would be a paper that I will look into in the future for data resources.

Minor/representation:

— Remind the reader that GBM denotes gradient boosted decision tree already in Figure 1, where it’s used for the first time.
— The abbreviation CPS is used the first time on page 3 but only introduced later on page 5.



**Time Spent Reviewing:**

3.5

---

> ### Author Response · Authors · 2021-08-10
> **Author response**
>
> We thank the reviewer for their comments and feedback. The reviewer’s primary concern is whether the paper’s contributions of new datasets and explication of empirical phenomena arising in algorithmic fairness are in-scope for NeurIPS and whether NeurIPS is the correct venue for this work. In recent years, there have been a number of important NeurIPS and ICML papers which introduce new datasets and examine empirical phenomena, but do not introduce new methods or prove new theorems. Among these include:
> - Barbu, Andrei, et al. "ObjectNet: A large-scale bias-controlled dataset for pushing the limits of object recognition models." Advances in Neural Information Processing Systems 32 (2019): 9453-9463.
> - Recht, Benjamin, et al. "Do imagenet classifiers generalize to imagenet?." International Conference on Machine Learning. PMLR, 2019.
> - Koh, Pang Wei, et al. "Wilds: A benchmark of in-the-wild distribution shifts." International Conference on Machine Learning. PMLR, 2021.
> - Piergiovanni, A. J., and Ryoo, Michael. "Avid dataset: Anonymized videos from diverse countries." Advances in Neural Information Processing Systems 33 (2020).
> - Veillette, Mark, et al. "SEVIR: A Storm Event Imagery Dataset for Deep Learning Applications in Radar and Satellite Meteorology." Advances in Neural Information Processing Systems 33 (2020).
> - Hu, Weihua, et al. "Open Graph Benchmark: Datasets for Machine Learning on Graphs." Neural Information Processing Systems (NeurIPS) (2020).
>
> In algorithmic fairness in particular, there have been several calls for precisely this type of dataset and empirical work, including [1, 2, 3, 4]. Our paper makes contributions along both of these lines. As Reviewer 2 writes, our paper “addresses...the complete lack of high-quality datasets” in the algorithmic fairness space. Moreover, the empirical phenomena we expose offers several new directions for methods work, both for algorithmic fairness and for tabular data more broadly.
>
> Consequently, we view the contributions of our paper in line with these papers (and many others) which have previously appeared at NeurIPS and similar venues.
>
> [1] Gebru, Timnit, et al. "Datasheets for datasets." arXiv preprint arXiv:1803.09010 (2018).
> [2] Jo, Eun Seo, and Timnit Gebru. "Lessons from archives: Strategies for collecting sociocultural data in machine learning." Proceedings of the 2020 Conference on Fairness, Accountability, and Transparency. 2020.
> [3] Paullada, Amandalynne, et al. "Data and its (dis) contents: A survey of dataset development and use in machine learning research." arXiv preprint arXiv:2012.05345 (2020).
> [4] Sambasivan, Nithya, et al. "“Everyone wants to do the model work, not the data work”: Data Cascades in High-Stakes AI." proceedings of the 2021 CHI Conference on Human Factors in Computing Systems. 2021.

---

### Decision · Program_Chairs · 2021-09-27

**Decision:**

Accept (Oral)

**Comment:**

Thanks for the strong submission. The reviewers were unanimous that the paper provided a valuable contribution, and disagreed only on whether it was submitted to the right track. Our job is to provide platform for good work, and since there was no disagreement on the quality of the work, we decided to accept.